

# *Trichodesmium* physiological ecology and phosphate reduction in the western Tropical South Pacific

Kyle R. Frischkorn[1,2], Andreas Krupke[3*], Mónica Rouco[2**], Andrés E. Salazar Estrada[1,2], Benjamin A. S. Van Mooy[3], and Sonya T. Dyhrman[1,2***]

[1]Department of Earth and Environmental Sciences, Columbia University, New York, NY, U.S.A.
[2]Lamont-Doherty Earth Observatory, Columbia University, Palisades, NY, U.S.A.
[3]Department of Marine Chemistry and Geochemistry, Woods Hole Oceanographic Institution, Woods Hole, MA, U.S.A.

*Present address: Thermo Fisher Scientific, Life Science Solutions, 180 Oyster Point Blvd., South San Francisco, CA, U.S.A
**Present address: K=1 Project, Center for Nuclear Studies, Columbia University, New York, NY, U.S.A.
****Correspondence to*: Sonya T. Dyhrman (sdyhrman@ldeo.columbia.edu)

## Abstract

$N_2$ fixation by the genus *Trichodesmium* is predicted to support a large proportion of the primary

productivity across the oligotrophic oceans, regions that are considered among the largest biomes on

Earth. Many of these environments remain poorly sampled, limiting our understanding of

*Trichodesmium* physiological ecology in these critical oligotrophic regions. *Trichodesmium* colonies,

communities that consisted of the *Trichodesmium* host and their associated microbiome, were collected

across the oligotrophic western tropical South Pacific (WTSP). These samples were used to assess host

clade distribution, host and microbiome (holobiont) metabolic potential, and functional gene expression,

with a focus on identifying *Trichodesmium* physiological ecology in this region. Expression dynamics

across the WTSP transect indicated potential co-limitation of *Trichodesmium* for phosphorus and iron.

A gene cassette for phosphonate biosynthesis was detected in *Trichodesmium,* the expression of which

co-varied with the abundance of *Trichodesmium* Clade III*,* which was unusually abundant relative to

Clade I in this environment*.* Coincident with the expression of the gene cassette, phosphate reduction

to phosphite and low molecular weight phosphonate compounds was measured in *Trichodesmium*

colonies as well as genes that enable use of this reduced phosphorus in both *Trichodesmium* and the

microbiome. Overall, these results highlight physiological strategies for survival by the *Trichodesmium*

holobiont in the oligotrophic ocean, revealing mechanisms with the potential to influence the cycling of

resources like nitrogen and phosphorus.



## 1 Introduction

The oligotrophic oceans extend over approximately 70% of Earth and are characterized by chronically low nutrient concentrations that limit primary productivity in these biomes (Moore et al., 2013). Within oligotrophic marine environments, $N_2$ fixing microorganisms can serve as a source of "new" nitrogen

that is bioavailable to other organisms. Among these marine diazotrophs, the colonial, filamentous cyanobacterium *Trichodesmium* plays a disproportionately large role in the cycling of carbon, phosphorus and nitrogen: it supplies fixed carbon through photosynthesis, was recently found to be a hotspot of phosphate reduction (Van Mooy et al., 2015), and has been estimated to be responsible for approximately half of the biologically fixed $N_2$ in the ocean (Bergman et al., 2013; Capone et al., 1997).

As such, the efficiency of the biological pump in sequestering carbon in the deep ocean is dependent in part on the distribution and activities of diazotrophic organisms like *Trichodesmium,* and an understanding of how this organism's physiology and ecology varies across diverse environments is a critical aspect of understanding present, and future, global biogeochemical cycles.

Diazotrophy frees *Trichodesmium* from nutrient limitation by nitrogen. As such, the distribution and activities of this cyanobacterium are predominantly influenced by the availability of phosphorus and iron in the surface ocean, which vary depending on the ocean basin and its proximity to supply of these resources (Moore et al., 2013; Sohm et al., 2011). Evidence of the intense competition for phosphorus and iron is evident in the suite of physiological strategies that this organism is known to employ. These

strategies include the production of transporters and enzymes that take up and hydrolyze diverse organic and reduced phosphorus compounds (Dyhrman et al., 2006; Orchard et al., 2009; Polyviou et al., 2015), or enable the uptake and storage of organic and inorganic iron (Polyviou et al., 2018; Snow et al., 2015). The genes encoding these functions are expressed *in situ* across diverse environments, indicating that competition for these resources is a critical aspect of *Trichodesmium* physiology (Chappell et al., 2012;

Dyhrman et al., 2006; Rouco et al., 2018). Recent evidence from culture studies also suggests that *Trichodesmium* employs a unique set of physiological strategies to cope with co-limitation of phosphorus and iron that differs from that of either resource alone (Walworth et al., 2017). This response is, in part, characterized by the down-regulation of $N_2$ fixation in favour of uptake and




hydrolysis of organic nitrogen sources, which could influence *Trichodesmium's* capacity to fuel primary productivity if these strategies are employed in the oligotrophic ocean (Walworth et al., 2017).

Survival in oligotrophic environments might also be enabled by biological interactions within
*Trichodesmium* colonies. *Trichodesmium* has long been known to occur with tightly associated bacteria that are unique from those free-living in the water column (Hmelo et al., 2012; Paerl et al., 1989). Recent evidence suggests that these interactions are ubiquitous and taxonomically conserved (Lee et al., 2017; Rouco et al., 2016) and that this epibiotic bacterial community, referred to as the *Trichodesmium* microbiome, contains a large amount of metabolic potential that exceeds and complements that of the
*Trichodesmium* host in populations from the western North Atlantic (Frischkorn et al., 2017). Coordinated gene expression patterns within the holobiont (*Trichodesmium* and its microbiome) suggest an interdependence of the microbiome on host-derived fixed carbon, $N_2$ and vitamins, and suggests microbiome respiration could create conditions that favour continued diazotrophy and photosynthesis (Frischkorn et al., 2018; Paerl and Bebout, 1988). The stability of these relationships in the future ocean
is unknown, but they are likely to change. For example, incubation of cultured *Trichodesmium* colonies with an elevated carbon dioxide concentration resulted in significant changes in microbiome nutritional physiology (Lee et al., 2018). These microbiome changes have the potential to alter the amount of fixed $N_2$ and carbon that transfer from the colony to the environment at large. Overall, the continued appreciation of the importance of the microbiome in *Trichodesmium* ecology underscores that
investigations must consider these microbial communities as a holobiont in order to fully understand and predict their role in the future environment.

Geochemical drivers of *Trichodesmium* distribution and $N_2$ fixation are increasingly well characterized in regions of the ocean where either phosphorus or iron are limiting such as the North Atlantic and
North Pacific Subtropical Gyre (Rouco et al., 2018; Sañudo-Wilhelmy et al., 2001; Sohm et al., 2011). The western tropical South Pacific (WTSP) represents and understudied region of the world's oceans (Bonnet et al., 2018) with conditions characterized by chronically low concentrations of both iron and phosphate (Moore et al., 2013; Sohm et al., 2011). Despite low resources, this region can support high



levels of $N_2$ fixation, with rates exceeding 700 umol $m^{-2}$ $d^{-1}$ where this resource transfers across diverse ecological groups and ultimately supplies up 90% of to the photic zone with new nitrogen (Bonnet et al., 2017; Caffin et al., 2017, 2018). In this study, metagenomic and metatranscriptomic sequencing was leveraged along with taxonomic distribution, physiological activities and geochemical measurements to

better understand *Trichodesmium* physiological ecology in an under-sampled but important region of the oligotrophic ocean.

## 2 Materials and Methods

### 2.1 Biogeochemical analyses

Samples were collected across a transect of the western Tropical South Pacific (WTSP) during the OUTPACE cruise (Oligotrophy to UlTra-oligotrophy PACific Experiment, DOI: http://dx.doi.org/10.17600/15000900) aboard the *R/V L'Atalante* during austral summer (February – April) of 2015 (Fig. 1, Moutin *et al.*, 2017). Nutrient analyses were collected using a Rosette sampling

device and analysed as previously reported for the OUTPACE research expedition (e.g. Stenegren et al., 2017). Geochemical data are archived online on the LEFE CYBER Database (http://www.obs-vlfr.fr/proof/php/outpace/outpace.php). Data were visualized and contoured using Ocean Data View 4.6.2 with the DIVA grid method (R. Schlitzer; http://odv.awi.de).

Water column phosphate uptake rate was determined as described by Van Mooy et al., (2015), and is briefly outlined here. First, 50 mL aliquots of whole seawater collected in Niskin bottles were decanted into acid-washed polycarbonate vials. Next, 1 µCi of carrier-free [33]P-phosphoric acid was added to the bottles, which represented an amendment of approximately 10 pmol $L^{-1}$ of phosphoric acid. Then, bottles were incubated in an on-deck incubator for 2-4 hr. Finally, the seawater in the bottles was

filtered through a 25 mm diameter, 0.2 µm poresize polycarbonate membrane, and the radioactivity of the membrane was determined by liquid scintillation counting.





## 2.2 *Trichodesmium* clade sampling and analysis

Samples for *Trichodesmium* clade distribution analysis were obtained at select short duration stations across the transect from water depths ranging from 5 m to ~150 m using 12 L of water for each depth obtained from a Rosette sampling device filtered through 47 mm 10 μm pore size polycarbonate filters.

Filters were flash frozen and stored in liquid nitrogen until processing. DNA was extracted from filters and the abundance of *Trichodesmium* clade I (which encompasses *T. thiebautii, T. tenue, T. hildebrandtii,* and *T. spiralis*) and III (which encompasses *T. erythraeum* and *T. contortum*), which are considered to be the dominant *Trichodesmium* clades in the field (Rouco et al., 2014), was determined using quantitative polymerase chain reaction (qPRC) targeting the *rnpB* gene, following protocols

previously described (Rouco et al., 2014). Clade distribution data was displayed and contoured using Ocean Data View 4.6.2 with the DIVA grid method (R. Schlitzer; http://odv.awi.de).

## 2.3 *Trichodesmium* colony sampling

*Trichodesmium* colonies were sampled across the transect at approximately the same time (between 8AM and 10:50AM local time). *Trichodesmium* samples were obtained with six hauls of a 130 μm

mesh size net towed through surface sea water. Colonies were skimmed with a Pasteur pipette from the surface layer of net towed samples and then washed two times with 0.2 μm sterile-filtered surface seawater on 5 μm, 47 mm polycarbonate filters with gentle vacuuming to remove non-tightly associated microorganisms. All colony samples were cleaned and processed within 15 minutes of collection. Filters with colonies were flash frozen and stored in liquid nitrogen for DNA or RNA extraction.

## 2.4 Phosphate reduction in *Trichodesmium* colonies

Phosphate uptake and synthesis rates of low-molecular-weight (LMW) reduced phosphate (P(+3)) compounds in *Trichodesmium* colonies were determined as described by Van Mooy et al., (2015). Briefly, phosphate uptake by *Trichodesmium* colonies was determined by filling acid-washed

polycarbonate 50 mL bottles with filter-sterilized surface seawater and approximately 20 *Trichodesmium* colonies. Incubation in on-deck incubators and measurement by liquid scintillation counting proceeded as previously described for the whole water analyses. In parallel, to measure the





synthesis rates of LMW P(+3) compounds, after some sets of *Trichodesmium* incubations the filtered colonies were not immediately measured by liquid scintillation counting. Instead, the colonies were placed in a cryovial containing 1 mL of pure water and flash frozen in liquid nitrogen. These samples were then transported to the lab ashore and subjected to numerous freeze-thaw cycles to extract

intracellular LMW P(+3) compounds. The LMW P(+3) compounds in the extracts were then isolated by preparative anion chromatography. Two fractions were collected in retention time windows consistent with retention times of pure standards of 1) methylphosphonic acid, 2-hydroxy ethylphosphonic acid, and 2-amino ethylphosphonic acid; and 2) phosphorous acid. The $^{33}$P radioactivity in these two operationally-defined fractions is ascribed to LMW phosphonates and

phosphite, respectively.

### 2.5 DNA extraction and metagenome sequencing

Genomic DNA was extracted from *Trichodesmium* colony samples obtained from LDA, LDB and SD5 (Fig. 1) using the MoBio Power Plant Pro DNA Isolation Kit (MoBio Laboratories, Inc., Carlsbad, CA,

USA) following the manufacturer instructions. Genomic DNA extracts were sequenced at the Argonne National Lab (Lemont, IL, USA) following a *Trichodesmium* consortium protocol previously described (Frischkorn et al., 2017). Briefly, DNA was sheared with a Covaris Sonicator (Woburn, MA, USA), transformed into libraries with WaferGen Apollo324 automated library system (Clonetech Laboratories, Mountain View, CA, USA) and Illumina compatible PrepX ILMN DNA kits (San Diego, CA, USA)

following manufacturer instructions. An average insert size of ~750 base pairs was targeted. Sage BluePippin (Beverly, MA, USA) was used to size select libraries prior to sequencing all three samples split across one 2 x 100 bp lane of the Illumina HiSeq2000. Metagenomic reads from these three samples are available on the NCBI Sequence Read Archive under BioProject number PRJNA435427.

### 2.6 Metagenomic sequence assembly and analysis

Raw reads were trimmed assembled into scaffolds and subsequently analysed following the protocol previously reported in Frischkorn et al., 2017. Briefly, reads were trimmed with Sickle (https://github.com/najoshi/ sickle), converted into fasta format, merged together and co-assembled with

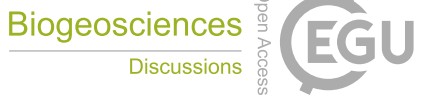

IDBA-UD (Peng et al., 2012) so as to create a South Pacific *Trichodesmium* holobiont genomic template to which future metatranscriptomic reads could be mapped. Assembled scaffolds were partitioned between *Trichodesmium* and heterotrophic bacteria (hereafter referred to as the microbiome) and functionally annotated after clustering into genome bins using MaxBin 2.0 with default parameters

(Wu et al., 2015). Taxonomic partitioning of binned scaffolds was carried out after translation of each scaffold into protein coding genes with the metagenomic setting of Prodigal (Hyatt et al., 2010), annotation of resultant proteins with the blastp program of DIAMOND (Buchfink et al., 2015) against the NCBI nr database, and classification with MEGAN6 (Huson et al., 2013) based on the phylogenetic classification of the majority of proteins within a genome bin. Functional annotations for translated

proteins were obtained by DIAMOND against the UniRef90 database (Suzek et al., 2007) with an e-value cut-off of $1\times10^{-3}$. Functional annotation was also carried out using the Kyoto Encyclopedia of Genes and Genomes (KEGG) with the online Automatic Annotation Server using the bi-directional best-hit method, the GHOSTX search program, and the prokaryote representative gene set options. Proteins from the merged assembly were also clustered into gene families of similar function or

orthologous groups (OGs) as previously reported (Frischkorn et al., 2017). Briefly, DIAMOND reciprocal blastp program results were processed using the program MCL (Markov cluster algorithm) set to an inflation parameter or 1.4. Consensus UniRef and KEGG annotations for each OG represent the majority annotation for all successfully annotated proteins within that group. Homologs to the phosphonate biosynthesis gene phosphoenolpyruvate phosphomutase (*ppm*) were found by screening

against manually annotated and reviewed Ppm proteins from the Swiss-Prot database. These verified proteins were aligned with Muscle version 3.8.425 with default parameters (Edgar, 2004), converted into HMM profiles with hmmbuild and hmmpress and used as a the database for hmmsearch, all using HMMER version 3.1 (Eddy, 1995). This HMMER approach was used to screen proteins generated from this study, as well as a previously published *Trichodesmium* consortia metagenome assembly from the

western tropical North Atlantic Ocean (Frischkorn et al., 2017) and protein sequences derived from the genome sequence of *Trichodesmium erythraeum* IMS101 (https://genome.jgi.doe.gov/portal/TrieryIMS101_FD/TrieryIMS101_FD.info.html). Sequence alignments were visualized using Geneious version 11.0.3 (http://www.geneious.com/, Kearse et al.,





2012) and important residues were obtained from previous crystal structure analyses (Chen et al., 2006). A reassembly of *Trichodesmium*-identified bins was performed to lengthen scaffolds in an attempt to provide genomic context to the *Ppm*-containing scaffold. The subset of all metagenomic reads mapping to *Trichodesmium* bins was selected using Samtools (Li et al., 2009) and the subseq program in Seqtk

(https://github.com/lh3/seqtk) and then reassembled using IDBA-UD as described previously. Maximum likelihood analysis of Ppm and related proteins was performed using the FastTree plugin in Geneious with default settings (Price et al., 2010), following a protocol previously employed for the annotation of environmental Ppm proteins (Yu et al., 2013). The sequences used to generate the tree were composed of proteins with homology to the identified *Trichodesmium* Ppm genes, as well as

similar sequences pulled from the NCBI nr database after online blastp analysis. The Interactive Tree of Life program was used to edit phylogenetic trees (Letunic and Bork, 2016). Proteins were also screened for MpnS, a protein that produces methylphosphonate, using the same protocol described above.

**2.7 RNA extraction and metatranscriptomic sequencing**

Prokaryotic RNA was extracted and sequenced from samples obtained from SD2, SD6 and SD9 as well as three samples from LDB (Fig. 1) following a protocol described previously (Frischkorn et al., 2018). Briefly, the Qiagen RNeasy Mini Kit (Qiagen, Hildern, Germany) was used to extract total RNA following manufacturer instructions, with the addition of 5 minutes of bead beating with ~500 μL 0.5 mm zirconia/silica beads after addition of Buffer RLT. On column DNase digestion (RNase-free DNase

Kit, Qiagen) was performed. A MICROBEnrich Kit (ThermoFisher Scientific, Waltham, MA, USA) was used to enrich the prokaryotic RNA fraction and ribosomal RNA was removed with the Ribo-Zero Magnetic kit for bacteria (Illumina), both following manufacturer instructions. Concentration and integrity of mRNA was assessed using a BioAnalyzer and the RNA 600 Nano Kit (Agilent Technologies, Santa Clara, CA, USA). Library preparation and sequencing was performed at the JP

Sulzberger Genome Center at Columbia University. Libraries were generated with the Illumina TruSeq RNA sample preparation kit. An Illumina HiSeq was used to sequence 60 million paired end reads for each sample. Metatranscriptomic reads from these six samples are available on the NCBI Sequence Read Archive under BioProject number PRJNA435427



## 2.8 Metatranscriptomic sequence analysis

Metatranscriptomic reads were trimmed, normalized and mapped as previously described (Frischkorn et

al., 2018). Briefly, raw reads were pre-processed following the Eel Pond Protocol for mRNAseq (Brown et al., 2013). Cleaned reads were mapped with RSEM (paired-end and bowtie2 options selected) (Li and Dewey, 2011) to the protein coding regions of the metagenomic scaffolds previously partitioned across *Trichodesmium* and the microbiome. Read counts were summed separately for *Trichodesmium* and microbiome fractions for all genes in an OG. Comparisons of relative enrichment

across sets of nutrient responsive genes were made using Kolmogorov-Smirnov tests to examine the null hypothesis that the expression of gene sets at a given station did not deviate significantly from the average expression of that set across the transect. Prior to testing, tags per million expression values for each OG were normalized to the average abundance of that OG across the six samples. This normalization equalized the relative contribution of individual OGs to the gene set as a whole, thereby

avoiding bias caused by highly expressed individual OGs. *P* values less than 0.05 were considered significant. Pairwise correlation coefficients between clade abundance, geochemical parameters and OG expression (in tags per million, TPM) were calculated using the cor function in R.

## 3 Results

### 3.1 Biogeochemistry

Across the study transect the phosphate concentration in surface water (0 - <30 m depth) ranged between 0.006 μmol $L^{-1}$ and 0.2 μmol $L^{-1}$ and in whole averaged 0.05 μmol $L^{-1}$. The phosphate turnover time in the water column microbial community was variable across the transect ranging between

approximately 2 hours and 800 hours (Fig. 1), averaging ~220 hours across all stations sampled. The water column phosphate uptake rate was similarly variable, ranging from $9.9 \times 10^{-6}$ μmol $L^{-1}$ $hr^{-1}$ to 0.08 μmol $L^{-1}$ $hr^{-1}$ with lower uptake measured at stations where turnover time was high and vice versa (Fig. 1). Iron concentrations (Fig. 1) averaged 0.5 nmol $L^{-1}$. Of the stations sampled for metatranscriptome





analyses, stations LDB1-3 had the highest concentration of both phosphate and iron, with average values of 0.03 μmol L$^{-1}$ and 0.65 nmol L$^{-1}$ respectively (Fig. 1). A summary of the geochemical parameters measured across the stations is included in Supplemental Table 1.

**3.2 *Trichodesmium* clade distribution**

*Trichodesmium* (combined cell counts of Clade I and Clade III) was detected at each station and at each depth sampled across the transect with an average concentration of ~37,000 cells L$^{-1}$ in the surface ocean (<5 m depth) and a minimum average concentration of ~56 cells L$^{-1}$ deeper than 100 m (Fig. 2a). Across the transect the average concentration was highest in the surface ocean (~43,000 cells L$^{-1}$) at
approximately 30 m depth. Overall, abundance was greater at the stations in the western half of the transect (stations west of 170°W and LDB) with cell concentrations between ~9,000 – 58,000 cells mL$^{-1}$ with an average of ~29,000 cells mL$^{-1}$ across all depths measured (Fig. 2a). Abundance decreased east of these stations concurrent with the transition into the subtropical gyre stations where average measured cell concentrations ranged from ~40 – 80 cells mL$^{-1}$ with an average of ~57 cells mL$^{-1}$ across
all depths measured (Fig. 2a). In addition to overall abundance of *Trichodesmium,* the contribution of Clade I (*T. thiebautii, T. tenue, T. hildebrandtii,* and *T. spiralis*) and Clade III (*T. erythraeum* and *T. contortum*) to the total *Trichodesmium* abundance at each station was determined (Fig. 2b). Average water column concentration of Clade I was the highest at SD6 (~47,000 cells mL$^{-1}$) and the lowest at SD15 (~41 cells mL$^{-1}$). Average water column concentration of Clade III was highest at SD2 (~13,000
cells mL$^{-1}$), lowest measured at SD9 (3,000 cells mL$^{-1}$), and below the limit of detection at SD14 and SD15. Across the transect, the *Trichodesmium* communities were dominated by Clade I which made up approximately 80% of the cells measured on average at each station, while Clade III made up approximately 20% on average (Fig. 2b). Though Clade I was dominant overall, at discrete stations and depths, generally in the surface ocean, the percentage of Clade III rose to nearly 50% of the measured
*Trichodesmium* community (Fig. 2b). Clade III was not detected in the subtropical gyre stations (SD14 and SD15).





### 3.3 Metagenomic characterization of WTSP *Trichodesmium* consortia

A merged metagenomic assembly of *Trichodesmium* consortia reads from selected stations across the OUTPACE transect (Fig. 1) yielded 801,858 scaffolds in total. Taxonomic binning partitioned scaffolds

into 48 genome bins with similar read coverage and tetranucleotide frequency. After phylogenetic analysis with MEGAN6, 18 of the bins were classified as *Trichodesmium,* 23 as heterotrophic bacteria, while the remaining bins had the majority of their proteins phylogenetically classified as Eukaryotes or other photosynthetic cyanobacteria. Subsequently, all *Trichodesmium*-identified scaffolds were merged and considered as the *Trichodesmium* fraction of the WTSP consortia sampled. Similarly, all

heterotrophic bacteria-identified bins were merged together and considered as the microbiome fraction. Together, these taxonomically verified scaffolds were translated into 198,156 proteins, which clustered into 75,530 gene families of putatively similar function, or orthologous groups (OGs). Within the WTSP consortia, *Trichodesmium* and their microbiome possessed 9,790 and 68,538 OGs respectively. The majority of these OGs were unique to the microbiome, with 2,798 (3.7%) of the total OGs

composed of proteins found in both the *Trichodesmium* and microbiome genome bins.

Functional annotation of OGs showed that the microbiome contained nearly 10 times more unique KEGG IDs (Supplemental Fig. 1a). The greatest differences in functional gene capacity between *Trichodesmium* and the microbiome were found in the Environmental Information Processing and

Carbohydrate and Lipid Metabolism modules (Supplemental Fig. 1b). Unique microbiome KEGG submodules included glycan, fatty acid, and carbohydrate metabolism functions. In the Environmental Information Processing category the microbiome possessed unique functions pertaining to peptide, nickel, phosphate, amino acid, and ABC transporters as well as an enrichment in proteins related to bacterial secretion systems (Supplemental Fig. 1b).

### 3.3 Global expression profiling of WTSP *Trichodesmium* consortia

A total of 7,251 *Trichodesmium* OGs recruited metatranscriptome reads from at least one sample, and a total of  21,529 microbiome OGs recruited metatranscriptome reads from at least one sample. Shifts in



*Trichodesmium* and the microbiome gene expression from station to station, and over time at the LDB station, were observed in a global analysis of expression using KEGG modules (Fig. 3). Hierarchical clustering of expression patterns in KEGG clustered OGs showed clear partitioning of functions from station to station in *Trichodesmium*. Such clear patterns were not mirrored in microbiome expression

across the stations sampled. *Trichodesmium* at SD2 and SD6 exhibited high expression of OGs related to the photosynthesis genes. *Trichodesmium* nitrogen metabolism functions exhibited enriched expression in the three LDB stations sampled, while the microbiome showed the opposite pattern with enriched nitrogen metabolism expression at the three SD stations (Fig. 3). Amino acid metabolism pathways had relatively increased expression at the SD stations in *Trichodesmium* and the microbiome

where nitrogen metabolism functions were lowest. Additionally, increased expression of KEGG functions in *Trichodesmium* related to lipid, vitamin, glycan, and lipopolysaccharide metabolism were mirrored to some extent in the microbiome at SD6. At station SD9, both *Trichodesmium* and the microbiome exhibited increased expression in KEGG transport systems for peptides, nickel, phosphate and amino acids, as well as the repair system (Fig. 3), a station with comparatively low phosphate

concentration and turnover time (Fig. 1). Summed counts (tags per million, TPM) for each OG in the *Trichodesmium* and microbiome fractions along with their most abundant UniRef annotations are provided in Supplemental Table 2.

**3.4 Expression related to phosphorus and iron physiology**

The expression of sets of *Trichodesmium* OGs known to be responsive to low nutrient conditions were assayed across the transect. The low P responsive set was composed of OGs previously shown to have increased relative expression under conditions of phosphate stress and included the alkaline phosphatases *phoA* (Tery_3467) and *phoX* (Tery_3845), the high affinity phosphate binding protein *sphX* (which is homologous to and clustered into one OG with *pstS*) (Tery_3534), a phosphate ABC

transporter (Tery_1967), phosphite dehydrogenase *ptxD* (Tery_0368) and the carbon-phosphorus lyase gene marker *phnJ* (Tery_5000) (Dyhrman et al., 2006; Orchard et al., 2009; Polyviou et al., 2015). The iron responsive set included OGs previously shown to have increased expression in experimental low iron cultures and *in situ* in low iron environments and included iron-binding cytochrome genes *petE* and



*petJ* (Tery_2563, Tery_2561), the ferric uptake regulators *fur1*, *fur2* (Tery_1958, Tery_3404) which clustered into the same OG, and *fur3* (Tery_1953), flavodoxins *fld1* and *fld2* (Tery_1666, Tery_2559), fructose bisphosphate aldolase class II *fbaA* (Tery_1687), the iron-stress induced protein *isiA* (Tery_1667), bacterioferritin (*bfr*, Tery_2787), and ferritin (*ftn*, Tery_2482) (Chappell et al., 2012; Chappell and Webb, 2010; Polyviou et al., 2018; Snow et al., 2015). Also assayed were a suite of OGs recently shown to be significantly enriched in cultures of *T. erythraeum* IMS101 following prolonged maintenance in co-limiting concentrations of phosphorus and iron (herein called the co-limitation responsive set) (Walworth et al., 2017). These OGs included the flavin-containing monoxygenase *FMO* (Tery_3826) that hydrolyzes organic nitrogen, 5-methyltetrahydropteroyltriglutamate—homocysteine methyltransferase *metE* (Tery_0847), the 3-dehydroquinate synthase *aroB* (Tery_2977), and beta-ketoacyl synthase (*OXSM*, Tery_3819 and Tery_3821 which clustered into one OG).

All OGs included in the three nutrient responsive *Trichodesmium* sets were expressed at each station. Furthermore, the OGs in these nutrient responsive sets had some of the highest expression levels detected across all *Trichodesmium* OGs (Fig. 4). In particular, the co-limitation marker genes and iron responsive genes have relatively high expression. The *metE* (co-limitation) and *isiA* (iron limitation) markers have the 3[rd] and 5[th] highest average expression overall, following behind a gene encoding a photosynthetic reaction centre protein and an uncharacterized gene (Fig. 4). Notably the *FMO* gene had nearly 10 times higher average relative expression in the three LDB samples relative to the SD samples, and the highest overall relative expression at LDB1 (Supplemental Table 2).

Changes in relative expression of the nutrient responsive OG sets were measured by performing Kolmogorov-Smirnov tests of expression levels at individual stations relative to the average expression of that OG set across all stations sampled. The signal of OGs in the low phosphorus marker set was significantly enriched in the second sample from LDB, and significantly lower in the third sample from LDB. The signal in the low iron marker set was significantly enriched at SD2, and significantly lower in the third sample from LDB. These significant deviations from the average in the single nutrient limitation sets did not overlap with samples where the co-limitation marker set deviated significantly



from the average. The co-limitation marker gene set was only found to be significantly enriched in the first sample from LDB (Fig. 4).

### 3.5 Evidence of phosphonate biosynthesis in *Trichodesmium*

A ~7,400 bp scaffold containing a cassette of genes encoding a phosphonate biosynthesis pathway was found in the *Trichodesmium* partitioned metagenomic scaffolds (Fig. 5a). This scaffold contained 6 protein coding regions. The first three genes were annotated as phosphoenolpyruvate phosphomutase (*ppm*), phosphonopyruvate decarboxylase (*ppd*), and 2-aminoethylphosphonate-pyruvate transaminase (*2-AEP-TA*). These first two genes are most similar to homologs in the UniRef database that belong to

the non-heterocystous diazotroph *Planktothrix agardhii, a* member of the order Oscillatoriales along with *Trichodesmium.* The *2-AEP-TA* is homologous to a gene in a Gammaproteobacterial *Beggiatoa sp.* The fourth, fifth and sixth genes in this scaffold were annotated as a methyltransferase, a cytidylyltransferase (both homologs to genes in *P. agardhii*), and a group 1 glycosyl transferase (homologous to a gene from a *Tolypothrix* sp., a freshwater cyanobacterium in the order Nostocales)

(Fig. 5a).

In order to verify genome binning of the *ppm*-containing scaffold into the *Trichodesmium* fraction, reassembly was attempted to lengthen this scaffold. This effort extended the length of the scaffold upstream of the *ppm* cassette and resulted in the addition of a 44 bp fragment at the 5' end with 95%

homology to a non-coding region in the *T. erythraeum* IMS101 genome, as well as a protein coding gene fragment with no known annotation (Fig. 5a).

Comparison of the putative *Trichodesmium ppm* gene's amino acid sequence against experimentally verified phosphonate bond forming enzymes in other organisms showed high sequence identity

(Supplemental Fig. 2). Across the length of the peptide sequence, the *Trichodesmium* Ppm was 72% and 66% similar to sequences from the freshwater cyanobacteria *Planktothrix sp.* and *Moorea producens*, 65% similar to that of the freshwater ciliate *Tetrahymena,* 63% similar to the blue mussel *Mytilus edulis*, and 35% similar to that of a bacterium in the *Streptomyces* genus. Furthermore, closer inspection





of the sequences of all organisms showed 100% identity across residues involved in cofactor and substrate interactions, as well as strong conservation across other key residues involved in the enzyme's tertiary structure as determined by crystal structure analysis and comparison to similar enzymes (Supplemental Fig. 2) (Chen et al., 2006). Phylogenetic analysis of *Trichodesmium* Ppm placed it in a

phylogenetic branch along with sequences from other $N_2$ fixing freshwater cyanobacteria (Supplemental Fig. 3).

A metagenome derived from *Trichodesmium* consortia in the western tropical North Atlantic (Frischkorn et al., 2017) was also re-screened for proteins with homology to the *Trichodesmium* Ppm

sequence from this dataset. Five homologous sequences were recovered from this North Atlantic metagenome, with two proteins falling in the branch of verified phosphonate producing Ppm proteins. One protein was located on a branch adjacent to the *Trichodesmium* Ppm sequences recovered from this South Pacific dataset (Supplemental Fig. 3). With the exception of the closely related North Atlantic sequence adjacent to the Ppm sequence recovered from this dataset, the other similar proteins from the

South Pacific and North Atlantic consortia did not exhibit amino acid identity at the key conserved residues determined from crystal structure analysis (Chen et al., 2006).

The *T. erythraeum* IMS101 genome was also screened for homologs to the Ppm protein recovered here, but no sequences with conserved amino acid identity across important conserved residues were

detected. Furthermore, another gene responsible for biosynthesis of methylphosphonate, MpnS, was not detected in proteins from the *Trichodesmium* or microbiome fraction.

All 6 genes in this scaffold recruited reads after metatranscriptome mapping of each sample, indicating that all genes in the phosphonate biosynthesis cassette were expressed at each station, with the

exception of the methyltransferase, which recruited no reads from LDB1 (Fig. 5b). Relative expression of *ppm* at the three SD stations and the average expression across the three LDB samples was significantly positively correlated with surface water column (~5-25 m) abundance of Clade III *Trichodesmium* (R = 0.98, *p* = 0.02, Pearson correlation). There was no correlation between *ppm*



relative expression and Clade I abundance ($p = 0.2$) or surface phosphate concentration ($p = 0.7$).
Coincident with expression of phosphonate biosynthesis genes at all stations sampled, phosphate
reduction by *Trichodesmium* colonies was also detected in each sample analysed (Fig. 5c). At the five
stations tested, approximately 2% of the radiolabeled phosphate taken up by colonies was reduced to
either a LMW phosphonate compound (methylphosphonate, phosphonoacetylaldehyde, or 2-
aminoethylphosphonate) or phosphite ($PO_3^{3-}$). Coincident with phosphonate production, both
*Trichodesmium* and the microbiome possessed and expressed markers for reduced phosphorus
metabolism, including the phosphonate (C-P) lyase *phnJ* (Tery_5000 in the *T. erythraeum* IMS101
genome) and *ptxD* (Tery_0368 in the *T. erythraeum* IMS101 genome)*, a gene identified as phosphite
dehydrogenase that is implicated in oxidation of phosphite to phosphate (Polyviou et al., 2015)
(Supplemental Table 2).

## 4 Discussion

### 4.1 *Trichodesmium* distributions in the oligotorphic WTSP

The WTSP is considered to be among the most oligotrophic environments in the global ocean due to
low concentrations of critical resources like nitrogen, phosphorus and iron, coupled with intense
stratification that prevents upwelling of remineralized nutrients (Moutin et al., 2008). In spite of
chronically depleted resources, a diverse assemblage of free-living and symbiotic diazotrophs thrive in
this region (Stenegren et al., 2017). In the WTSP, *Trichodesmium* is typically abundant, and the
*Trichodesmium* distribution determined here by clade-specific qPCR agreed with previous analyses
along this transect via genus specific qPCR that showed high abundance in the west, decreasing sharply
at the transition into the gyre (Stenegren et al., 2017). Low *Trichodesmium* relative abundance at the
easternmost stations (SD14 and 15) sampled along this transect were found despite high phosphate
concentrations detected in surface waters there. This could be due to the low iron concentrations at these
eastern stations limiting growth, and this is corroborated by the finding that *Trichodesmium* distribution
was positively correlated by both iron flux from nearby island sediments (Dupouy et al., 2018), and the





fact that rates of water column $N_2$ fixation were lowest in the easternmost stations along this transect (Bock et al., 2018). *Trichodesmium* biomass was too low to evaluate gene expression patterns for these easternmost stations, but regardless, these data are consistent with the importance of iron concentration as a driver of *Trichodesmium* distribution and activities in the WTSP. In addition to iron, phosphorus

also likely exerts a strong influence over *Trichodesmium* in the WTSP. The surface water concentration of phosphate measured along this transect was low in the context of the global ocean (Sohm et al., 2011 and references therein) and measured phosphate turnover times in the water column at some stations on the order of hours indicated there was intense competition for phosphate (Van Mooy et al., 2009). The WTSP is poorly sampled relative to other oligotrophic ocean basins (Bonnet et al., 2018; Luo et al.,

2012), and little is definitively known about the canonical resource controls that characterize this environment over prolonged time periods (Sohm et al., 2011). Herein, the *Trichodesmium* distributions are consistent with the potential roles of both iron and phosphorus in the physiological ecology of this genus in the WTSP.

**4.2 Expression of metabolic potential in WTSP *Trichodesmium* consortia**

 The *Trichodesmium*-identified metegenomic scaffolds from across this WTSP transect contained 9,790 OGs. This number exceed the 6,798 OGs found in a metagenomic survey of the western North Atlantic *Trichodesmium*, as well as the 2,982 OGs generated from the *T. erythraeum* IMS101 protein coding

genes (Frischkorn et al., 2017). This disparity could potentially be attributed to a greater diversity of *Trichodesmium* ecotypes in the WTSP that have a different genomic makeup than those found in the western North Atlantic. Additionally, this increase in gene families in field populations of *Trichodesmium* relative to the sequenced *T. erythraeum* isolate could reflect gains of function by horizontal gene transfer over time in the environment, a process that could act particularly strongly in

*Trichodesmium,* as this organism is not subjected to genome streamlining as is the case for other oligotrophic ocean bacteria (Walworth et al., 2015).




*Trichodesmium* does not exist in isolation, as filaments and colonies are associated with an assemblage of epibiotic microorganisms that co-occur ubiquitously in the environment (Lee et al., 2017; Rouco et al., 2016) and contribute a large amount of metabolic potential that could underpin success in oligotrophic, low nutrient environments (Frischkorn et al., 2017). Here, as in the western North Atlantic

(Frischkorn et al., 2017), the large amount of unique, microbiome-only OGs (65,740), the low amount of shared functions (2,798) between *Trichodesmium* and their associated epibionts, and the expression of diverse subsets of these OGs across the transect highlights the diversity and potential utility of metabolic potential contributed to the holobiont by the microbiome. At the broadest KEGG annotation level, the microbiome contained approximately 10 times the unique functions found within the

*Trichodesmium* fraction, and these functions were largely consistent with those observed previously in the western North Atlantic (Frischkorn et al., 2017). The presence of unique microbiome transporter functions, especially those related to the transport of phosphate and metals including iron, reflect the importance of these resources within the colony microenvironment that is likely depleted in these key resources. The enrichment of microbiome functions related to the transport and subsequent metabolism

of sugars, carbohydrates and lipids could reflect the transfer of fixed carbon from host to microbiome, as the genes encoding these functions are known to oscillate over day night cycles in lockstep with *Trichodesmium* photosynthesis and carbon fixation genes (Frischkorn et al., 2018). These oscillations may support respiration processes that help maintain an environment favourable for $N_2$ fixation, and the functional enrichment observed here could underpin interactions within the holobiont that help maintain

$N_2$ fixation in the WTSP, despite the scarcity of resources.

Across the transect broad shifts in OG expression from station to station underscore the tight coupling between *Trichodesmium* physiology and the local geochemical environment. These changes showed an increased relative expression of photosynthetic processes and KEGG functions in *Trichodesmium* that

are potentially indicative of growth and cell division in the western stations, followed by a transition to increased relative expression of nitrogen functions in the eastern stations. Though expression in the microbiome was overall more variable, an orthogonal pattern in nitrogen metabolism functions between *Trichodesmium* and the microbiome underscores the importance of biologically available nitrogen in the





holobiont community. Taken with the previously noted importance of microbiome respiration, these results further demonstrate how *Trichodesmium* ecology is not only influenced by geochemical parameters, but also by biological interactions.

### 4.3 *Trichodesmium* physiological strategies in low phosphorus and iron conditions

The expression of genes known to be markers of nutrient stress can be used to assess the physiology of microorganisms *in situ,* and thus infer geochemical drivers of physiological ecology. Genes responsive to low phosphorus and iron are well-studied in *Trichodesmium* (e.g. Chappell et al., 2012; Chappell and Webb, 2010; Dyhrman et al., 2006; Orchard et al., 2009; Snow et al. 2015) and a recent culture study assessed the physiological response of *Trichodesmium* to coupled low phosphate and low iron conditions, yielding a set of co-limitation marker genes (Walworth et al., 2017). Expression of many of these marker genes is not detectable at all in cultures grown under replete conditions (Chappell and Webb, 2010; Orchard et al., 2009) or in field samples with relatively high concentrations of resources like iron (e.g. Chappell et al., 2012). Overall, relative expression of marker OGs for low phosphorus, low iron and co-limitation was among the highest of all OGs in the dataset. Expression of these OGs was observed across all samples, indicating that there was intense scavenging of phosphorus and iron consistent with the resource limiting conditions in this oligotrophic environment. In the case of phosphorus, these expression signals were consistent with some stations with rapid phosphate turnover rates, and thus intense phosphate competition, observed in the community as a whole. These expression signals also suggest a switch to growing on phosphite, or phosphoester or phosphonate forms of dissolved organic phosphorus. Taken together these results strongly indicate the importance of phosphorus and iron as drivers for *Trichodesmium* growth and $N_2$ fixation in the WTSP. Many of the markers for co-limitation also had among the highest expression overall, underscoring potential co-limiting geochemical conditions across this transect. The relative expression patterns of these three OG sets did not typically significantly co-vary from station to station, suggesting that variations in resource limitation were not consistent across the transect. The co-limitation OG set was only significantly enriched above average expression at one station (LDB1), and this station did not concurrently feature significant enrichment in low phosphorus or iron sets, which indicates a partitioning in molecular



strategies where co-limitation patterns of expression are distinct from either iron or phosphate limitation individually. This is consistent with culture studies, where expression patterns related to iron and phosphate co-limitation were distinct in *T. erythraeum* (Walworth et al., 2017). The variation of iron and phosphate concentration observed across the study transect may variably drive this subsequent

enrichment of phosphate, iron, or co-limitation gene set expression.

One highly expressed co-limitation marker OG was identified as the flavin-containing monooxygenase (*FMO*), an enzyme that is predicted to oxidize the organic nitrogen compound trimethylamine which is common in the marine environment (Walworth et al., 2017). This OG had nearly 10 times higher

average relative expression in the three LDB samples relative to the SD samples, and the highest overall relative expression at LDB1. This enrichment was coincident with a crashing phytoplankton bloom concurrently detected at this station (De Verneil et al., 2017), which could be a source of organic matter to the water column. Together these results may indicate a shift towards increased metabolism of organic nitrogen, and decreased $N_2$ fixation in *Trichodesmium* under these conditions. The expression of

OGs encoding the enzymes of $N_2$ fixation were not clearly reduced at LDB1, so at this juncture it is difficult to know if these expression signals are indicative of co-limitation, a response to increases in organic nitrogen found during the bloom decline, or a combination of factors. Regardless, such apparent alterations in *Trichodesmium* physiology could have important implications for rates of $N_2$ fixation in this region of the WTSP, both at present and in future ocean conditions of increased $CO_2$ and enhanced

oligotrophy. More studies in this under-sampled region would help to further resolve drivers of *Trichodesmium* physiological ecology.

**4.4 Phosphonate biosynthesis by *Trichodesmium* in the WTSP**

Phosphate exists in vanishingly low concentrations in the oligotrophic surface ocean and the activity of diazotrophs increases the demand for phosphorus by relieving nitrogen stress—a process that is enhanced by periodic increases in iron availability (Moutin et al., 2005). In low phosphate environments marine microbes, like *Trichodesmium,* can hydrolyse phosphate from organically bound compounds



like phosphoesters and phosphonates, the concentration of which far surpasses phosphate in the oligotrophic ocean (Dyhrman et al., 2007). The production and hydrolysis of reduced compounds like phosphonates are of particular interest because the hydrolysis of methylphosphonate has the potential to release methane, a potent greenhouse gas (Karl et al., 2008; Repeta et al., 2016). Previous studies

showing biosynthesis of phosphonates by certain *Trichodesmium* isolates (Dyhrman et al., 2009) as well as rapid phosphate reduction to phosphonate and phosphite and release by *Trichodesmium* colonies in the environment (Van Mooy et al., 2015) implicate this diazotroph as an important player in phosphonate biogeochemistry, yet the molecular mechanisms underlying phosphonate biosynthesis are poorly understood for this genus.

A *Trichodesmium* scaffold containing the full set of genes necessary to synthesize phosphonate compounds was recovered from metagenomes assembled from this WTSP transect. The *Trichodesmium* origin of this scaffold is supported by tetranucleotide frequency and metagenomic read mapping coverage, as well as the presence of a stretch of non-coding DNA with homology to the *T. erythraeum*

IMS101 genome. Furthermore, the protein in this scaffold identified as phosphoenolpyruvate phosphomutase (Ppm), the enzyme that carries out the formation of a carbon-phosphorus bond using phosphoenolpyruvate as a substrate (McGrath et al., 2013), was phylogenetically most similar to Ppm sequences from cyanobacteria like *Planktothrix* that are closely related to *Trichodesmium*. The phylogenetic distance between the *Trichodesmium* Ppm and those of heterotrophic bacteria further

support that this scaffold was recovered from a *Trichodesmium* genome and not from a member of the microbiome.

The molecular machinery necessary to synthesize phosphonates is evolutionarily conserved and the biosynthesis of phosphonoacetaldehyde is the starting point from which a diverse suite of organic

phosphonate compounds can be produced (McGrath et al., 2013). Based on the genes in this *Trichodesmium* scaffold, synthesis begins with the formation of the carbon-phosphorus bond after molecular rearrangement of phosphoenolpyruvate to phosphonopyruvate, catalysed by Ppm. Next, phosphonopyruvate decarboxylase (Ppd), the protein encoded by the following gene in the cassette,





likely performs the irreversible conversion of phosphonopyruvate to phosphonoacetaldehyde which prevents reversion to the ester bond structure. Finally, the presence of the gene for 2-aminoethylphosphonic acid pyruvate-transaminase (*2-AEP-TA*) suggests that phosphonoacetaldehyde is further converted to 2-aminoethylphosphonate (2-AEP), the organophosphonate that occurs most

5  commonly in the environment (McGrath et al., 2013). The *mpnS* gene mediates the production of methylphosphonate down stream of *ppm* in the marine microbes where it has been detected (Metcalf et al., 2012). There was no evidence of *mpnS* in *Trichodesmium* or the microbiome, but *Trichodesmium*-derived phosphonates could potentially be further modified to methylphosphonate by organisms not associated with colonies.

The *ppm* gene can be found in approximately 7% of microbial genome equivalents recovered from the Global Ocean Survey, and of these *ppm*-containing genomes, 20.6% are estimated to be cyanobacterial in origin (Yu et al., 2013). A protein with homology to Ppm was previously detected and attributed to *Trichodesmium* in metagenomic samples from the western North Atlantic (Frischkorn et al., 2017),

15  though this gene was not found to be part of a 2-AEP synthesis cassette. The detection of these phosphonate biosynthesis genes in environmental *Trichodesmium* sequences corroborates and provides a potential mechanism that may in part underpin phosphate reduction rates measured in the western North Atlantic (Van Mooy et al., 2015). The genes on the *ppm*-containing *Trichodesmium* scaffold identified here recruited metatranscriptomic reads from each sample sequenced, suggesting active use of

20  these enzymes across the transect. Furthermore, reduced phosphonate compounds (which would include 2-AEP) were produced from radiolabeled phosphate taken up by *Trichodesmium* colonies at each station analysed. Together, these results clearly illustrate a pathway by which *Trichodesmium* synthesizes phosphonates, and that this pathway is active in the WTSP and likely other environments like the western North Atlantic.

The relative expression of the *ppm* gene across the transect was significantly correlated with the abundance of *Trichodesmium* Clade III. Although this is consistent with the detection of phosphonates in cultured *Trichodesmium* isolates from Clade III (Dyhrman et al., 2009), no clear homologs of *ppm*





have been found in the *T. erythraeum* IMS101 genome assembly. At this juncture, this may be attributable to an incomplete genome, or the pathway could be mediated with a different mechanism, and more work would be required to screen *Trichodesmium* isolates for *ppm*. Regardless, these field data are consistent with culture studies that suggest that phosphonate biosynthesis may be variably

present in different *Trichodesmium* clades or species. Across this WTSP transect, Clade III accounted for up to 47.5% of the total *Trichodesmium* population in some locations, and in general composed nearly 20% of the population on average from station to station. This percentage of Clade III is considerably higher than that observed in the western North Atlantic (Rouco et al., 2014) and higher than the North Pacific subtropical gyre (Rouco et al., 2016). If phosphonate biosynthesis is consistently

a unique feature of Clade III, then *Trichodesmium*-mediated phosphonate production could be higher in the South Pacific than other ocean basins. Furthermore, models project that future ocean conditions will select for the dominance of *T. erythraeum* (Clade III) over that of Clade I species like *T. thiebautii* which are most abundant in the present ocean (Hutchins et al., 2013). Such an expansion of *T. erythraeum* could lead to a subsequent increase in phosphonate biosynthesis and an increase in the

availability of these compounds in the water column.

The production of a recalcitrant form of phosphorus and its potential release into the oligotrophic environment could have important consequences for microbial communities. Phosphonates are a critical source of phosphorus in the oligotrophic ocean, and the ability to utilize this resource could influence

microbial ecology in low nutrient environments. Across this transect, both *Trichodesmium* and their microbiome contained and expressed genes related to phosphonate catabolism, including the marker of the C-P lyase enzyme complex, *phnJ*. *Trichodesmium* and microbiome *phnJ* genes have also been detected and expressed in *Trichodesmium* communities from the chronically low phosphate western North Atlantic ocean as well as the North Pacific subtropical gyre (Dyhrman et al., 2006; Frischkorn et

al., 2017, 2018). In addition to the production and hydrolysis of phosphonate compounds, we also detected evidence of the use of other forms of reduced phosphorus. The expression of the *ptxD* gene which is responsible for the oxidation of phosphite, another reduced phosphorus compound (Polyviou et al., 2015), was also expressed by *Trichodesmium* and the microbiome at all stations sampled. This



finding suggests active transformation and exchange of reduced phosphorus compounds between consortia members. In the low phosphorus western North Atlantic, up to 16% of the phosphate taken up by *Trichodesmium* colonies has been shown to be reduced and subsequently released from cells, an amount of phosphorus cycling that rivals the amount input to marine systems from allochthonous or

atmospheric sources (Van Mooy et al., 2015). The evidence of utilization of these traits across diverse geochemical environments, like the WTSP, and the large quantities of phosphorus they recycle suggests that phosphonate cycling composes an integral facet of the *Trichodesmium* holobiont's physiology, yet the reasons behind this cycling remain enigmatic. Not all microbes can metabolize phosphonates (Villarreal-Chiu, 2012), therefore it could also be plausible that formation of such compounds creates a

cryptic phosphorus pool that would in part restrict access to this critical nutrient by other microbes. In short, exchange of these compounds within the *Trichodesmium* holobiont, especially if through a cryptic pool, could help support $N_2$ fixation in *Trichodesmium* by modulating access to bioavailable phosphorus in the oligotrophic WTSP (Van Mooy et al., 2015).

Marine $N_2$ fixation is expected to increase in future oceans that are predicted to have higher temperatures and $CO_2$ concentration (Hutchins et al., 2007), and *Trichodesmium* cultures incubated in high $CO_2$ conditions exhibited irreversibly increased rates of $N_2$ fixation (Hutchins et al., 2015). In such conditions, cycling of phosphonate compounds that are not accessible to the full microbial community could support enhanced $N_2$ fixation, the release of new nitrogen into the water column and subsequently

fuel primary production. In future studies it will be important to assay how future ocean conditions will alter the clade distribution of *Trichodesmium* in the environment, as this could play a role in determining the potential flux of phosphonate compounds from colonies to the water column.

## 5 Conclusion

Marine microbes interact and alter the environment through abiotic transformations as well as through biotic interactions with one another and across trophic levels, and these processes work in tandem to influence global biogeochemical cycles. Understanding these processes *in situ* is of paramount importance to forecasting the ocean's role in the future climate, yet challenges persist with sampling





remote locations and filling knowledge gaps surrounding the ecology and physiology of key species. The OUTPACE research expedition afforded a unique opportunity to investigate communities of the keystone N$_2$ fixer *Trichodesmium* and their microbiome in the under-sampled South Pacific. Metagenomic and metatranscriptomic data showed a majority of unique physiological functions within

the microbiome, many of which were expressed in situ, and these functions may be important in maintaing *Trichodesmium* N$_2$ fixation in low resource environments like the WTSP. Patterns of iron, phosphate, and co-limitation gene set expression suggested that *Trichodesmium* may variably experience resource limitation across the transect, including potential co-limitation, which could modulate growth and N$_2$ fixation. A *Trichodesmium* gene cassette for the biosynthesis of the

phosphonates, its expression, and corresponding phosphate reduction rate measurements suggested that *Trichodesmium* is producing reduced phosphate in the WTSP. This finding expands the environments where phosphate reduction has been detected, and confirms the role of *Trichodesmium* in this poorly understood aspect of phosphorus biogeochemistry. Collectively, these data underscore the importance of iron and phosphorus, and the microbiome, in jointly driving the physiological ecology of this key

diazotroph in the WTSP

**Competing interests**

The authors declare they have no conflicts of interest.

**Acknowledgments**

This is a contribution to the OUTPACE (Oligotrophy to UlTra-oligotrophy PACific Experiment) project which was funded by the French national research agency (ANR-14-CE01-0007-01), the LEFE-CyBER program (CNRS- 8 INSU), the GOPS program (IRD) and the CNES (BC T23, ZBC 4500048836). The authors thank Thierry Moutin and Sophie Bonnet for leading the OUTPACE

expedition and for the invitation to participate. The authors also thank the OUTPACE science party and the captain and crew of the R/V *L'Atalante* for their assistance and camaraderie while underway. Special thanks to Andrea Caputo and Marcus Stenegren of Stockholm University for assistance while at sea. The authors also acknowledge the Carbonate cluster of the National Center for Genome Analysis



Support, and Carrie Ganote in partiulcar, for bioinformatics assistance. Grants from the National Science Foundation to STD (OCE-1332912) and BASVM (OCE-1536346 and OCE-1332898) supported this research. KRF is partially supported by a National Science Foundation Graduate Research Fellowship (DGE-16-44869).

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



## Figures.

**Figure 1.** Surface water (>30 m) phosphate concentration, iron concentration, community phosphate turnover and community phosphate uptake rate measured at stations across the OUTPACE (Oligotrophy to UlTra-oligotrophy PACific Experiment) transect during austral summer (February – April) of 2015. Numbers above and below indicate the short duration (denoted SD) or long duration (denoted LDA, LDB or LDC) stations where samples were obtained across the transect. In the top panel, * indicates stations where metagenomic or metatranscriptomic samples were obtained and † indicates stations where *Trichodesmium* clade distribution samples were obtained.

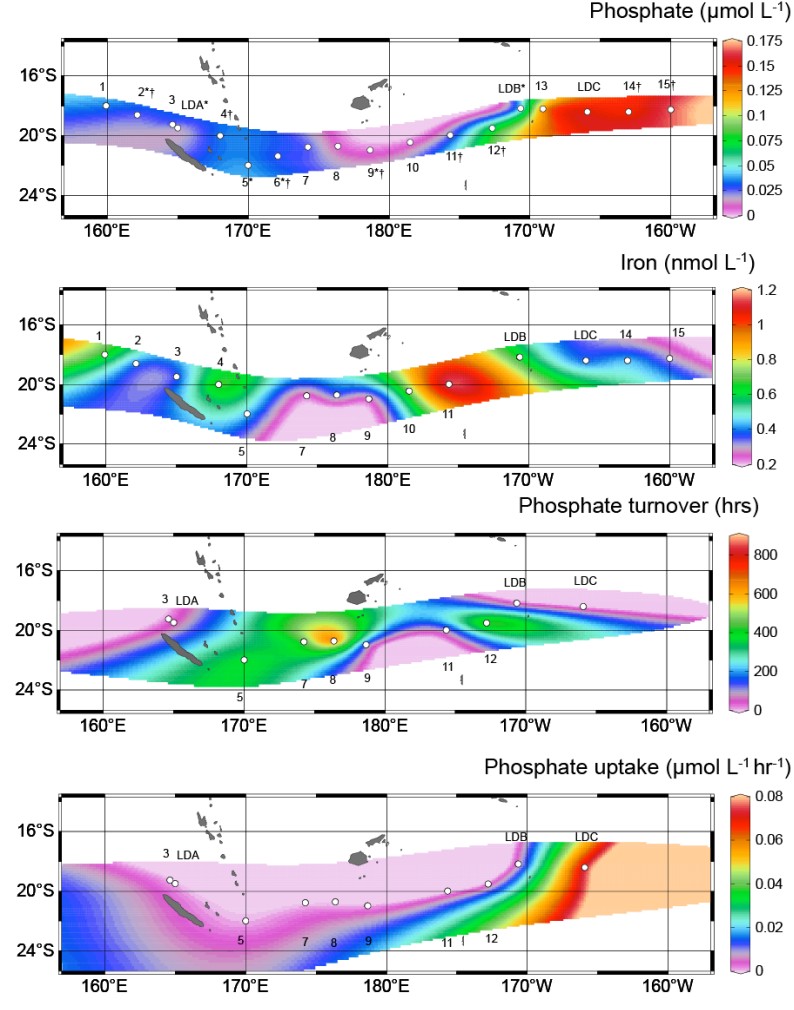



**Figure 2.** Abundance and clade distribution of *Trichodesmium*. Black dots denote the depths at which samples were taken, while station number is indicated above the panels. (a) Concentration of total *Trichodesmium* cells detected (cells L⁻¹). (b) Relative proportion of *Trichodesmium* Clade I (top panel) and Clade III (bottom panel) across this transect. Clade I includes *T. thiebautii, T. tenue, T.* 5 *hildebrandtii, and T. spiralis*. Clade III includes *T. erythraeum* and *T. contortum*. * indicates stations where metatranscriptomic sequences were sampled.

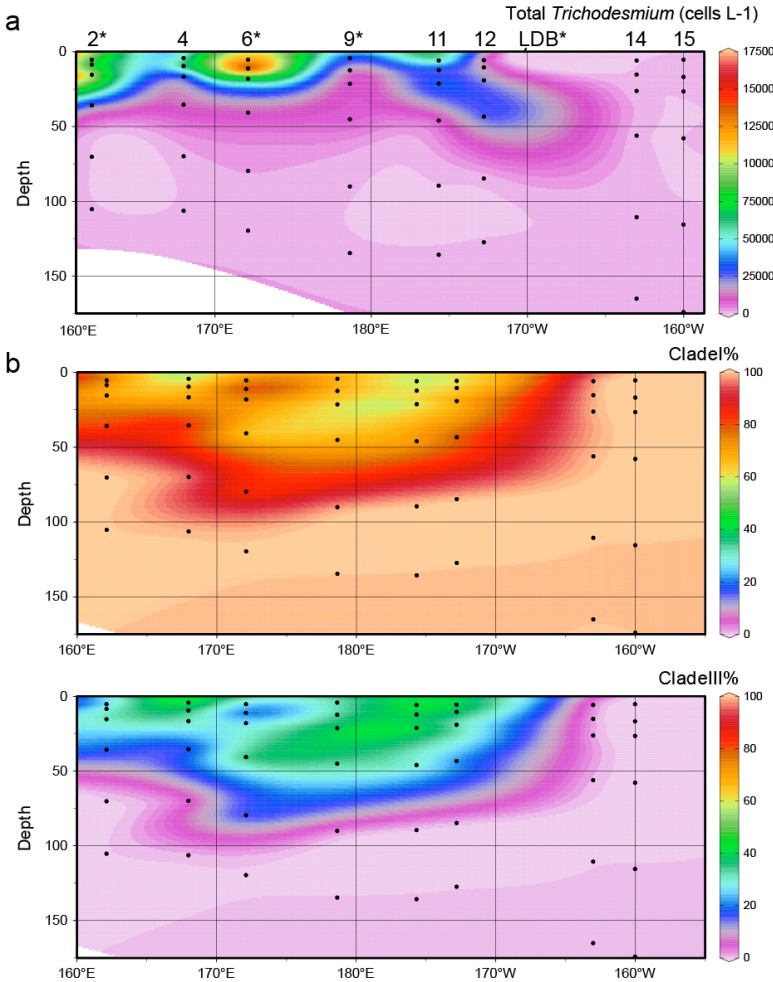





**Figure 3.** Heatmap of the summed relative expression of all OGs belonging to each KEGG category at short duration (SD) and long duration (LDB) stations, partitioned between *Trichodesmium* and the microbiome. Relative expression values represent row averages for either *Trichodesmium* or the microbiome for that particular OG. OGs were hierarchically clustered based on relative expression level. Note that no microbiome OGs were detected in the Metabolic capacity module.

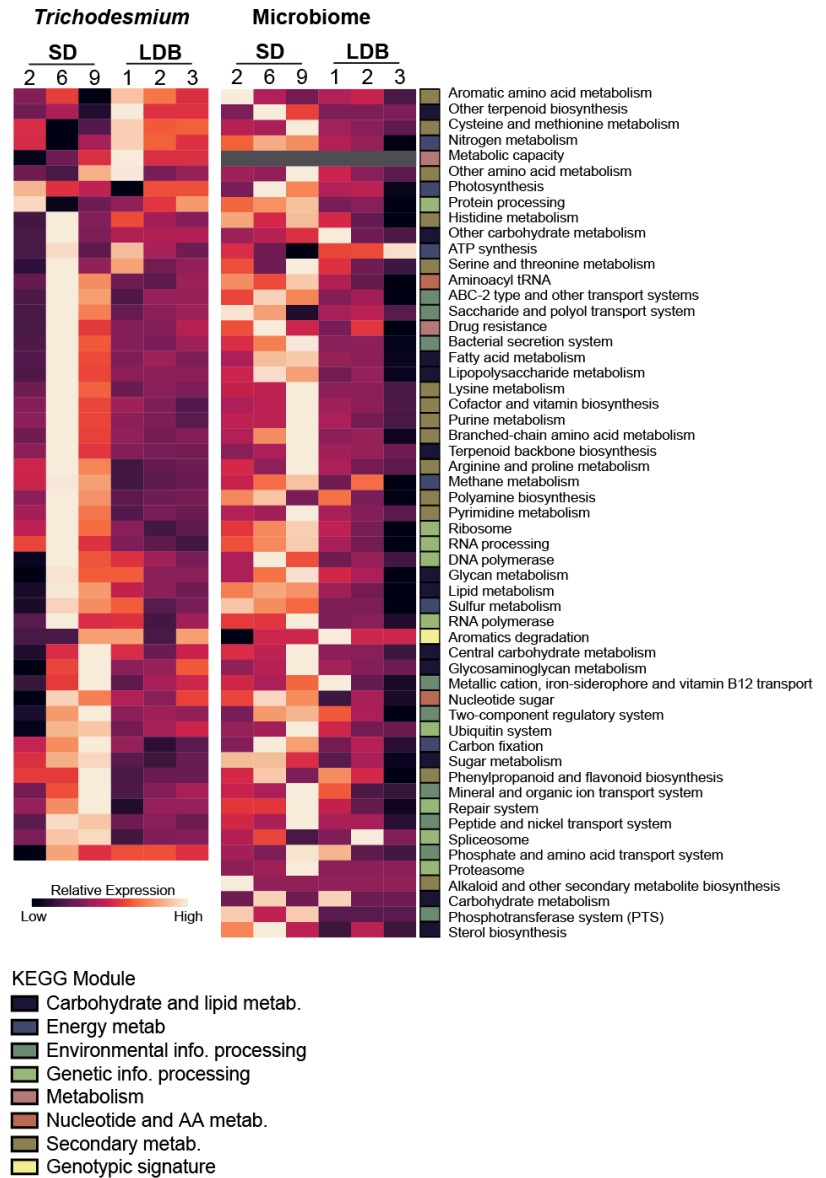



**Figure 4.** Expression patterns in nutrient responsive OG sets in *Trichodesmium*. (a) Average expression levels (in tags per million, TPM) for all OGs across all six stations ordered from highest to lowest expression, with OGs in the nutrient responsive sets highlighted. All other OGs appear grey. The bracket on the left panel demarks the region expanded in the panel on the right. (b) Distribution of expression patterns in gene sets known to be significantly responsive in *Trichodesmium* to low phosphorus (P), low iron (Fe), and P/Fe co-limiting conditions. Distributions for each set at each station were compared to the average distribution across all six stations using Kolmogorov-Smirnov tests to examine the null hypothesis that the expression of gene sets at a given station did not deviate significantly from the average expression of that set across the transect. Whiskers show the normalized enrichment level for the least and most enriched OG in that set. Boxes denote the upper and lower 25th percentiles, while the line indicates the median enrichment expression value. Asterisks indicate significance: * $p < 0.05$, ** $p < 0.005$, *** $p < 0.0005$, **** $p < 0.0001$. Black triangles denote whether significant stations were increased or decreased relative to the mean.

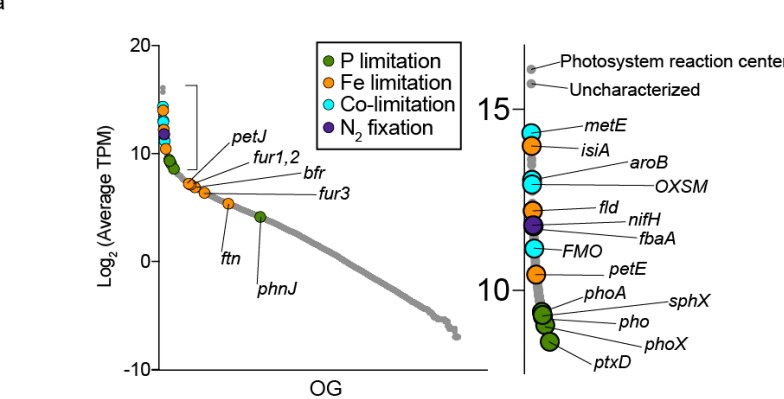

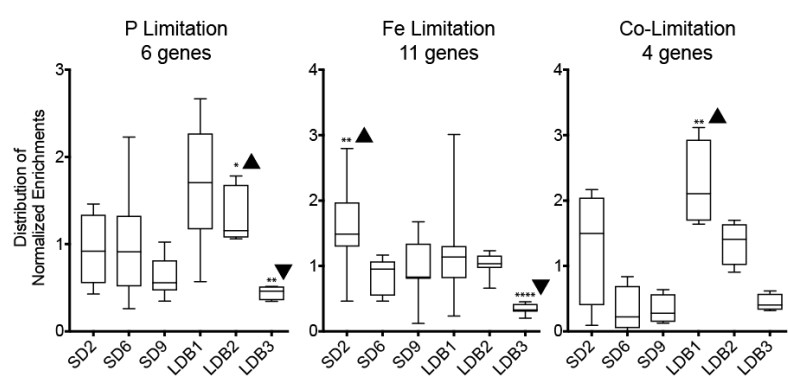



**Figure 5.** Annotations, topology and expression levels of a scaffold containing a phosphonate biosynthesis cassette recovered from a *Trichodesmium* identified metagenome bin. (a) Gene organization and annotations across the scaffold where arrows represent direction of transcription. * denotes a region with 95% homology to a non-coding region of the *T. erythraeum* IMS101 genome. (b) Normalized expression (tags per million, TPM) of each OG on this *ppm*-containing scaffold at each short duration (SD) or long duration (LDB) station sampled. Column colours match those of the top panel (a). (c) Percentage of total radiolabeled phosphate taken up by *Trichodesmium* colonies and reduced into low molecular weight (LMW) phosphonate compounds (including methylphosphonate, phosphonoacetylaldehyde, and 2-aminoethylphosphonate) and phosphite. Error bars denote standard deviation. Abbreviations: *ppm,* phosphoenolpyruvate phosphomutase; *ppd,* phosphonopyruvate decarboxylase; *2-AEP-TA,* 2-aminoethylphosphonate-pyruvate transaminase; *Me-T,* SAM-dependent methyltransferase; *Cy-T,* cytidylyltransferase; *GT,* group 1 glycosyltransferase.

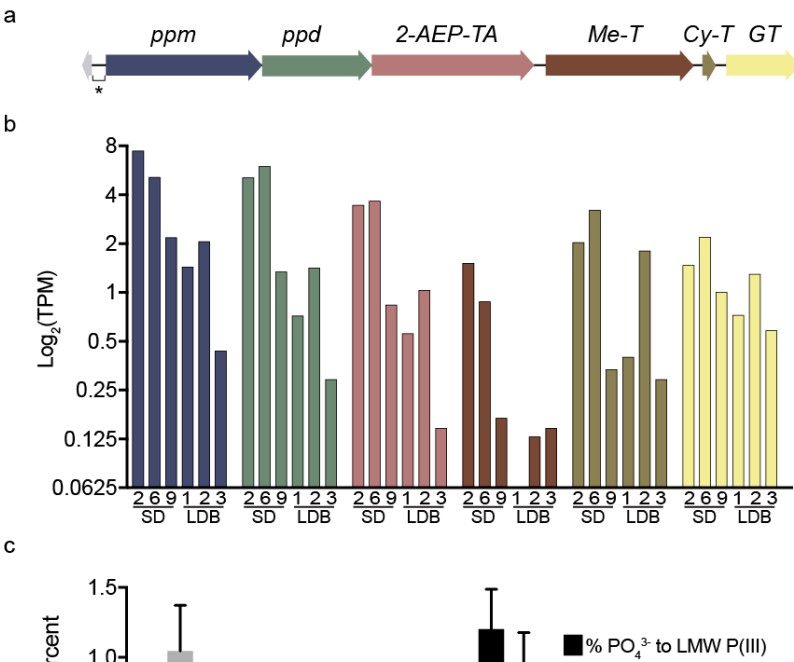