# Peer review of "Trichodesmium* physiological ecology and phosphate reduction in the western Tropical South Pacific"

_Biogeosciences, 2018_

## Referee Comment (RC1) · U. Pfreundt (Referee) · 23 Apr 2018

Referee comment
**Trichodesmium physiological ecology and phosphate reduction in the western Tropical South Pacific**

**General comments**

The article *Trichodesmium physiological ecology and phosphate reduction in the western Tropical South Pacific* by Frischkorn et al. is one of a series of articles on the scientific results of the OUTPACE project, a field study focused on the undersampled Western Tropical South Pacific. The authors used a combination of environmental metagenomics, metatranscriptomics, nutrient analyses, and analysis of *Trichodesmium* holobiont phosphate physiology to shed some light on the physiological interactions between and combined metabolic potential of *Trichodesmium* and its associated microbiome. The study focuses on phosphorus physiology, and touches on iron limitation as well.

I find the article generally very well written and easy to follow. The results presented are relevant to the understanding of how *Trichodesmium* and its microbiome shape nutrient availability in the regions where they thrive. Studies on *Trichodesmium* as a holobiont are being published increasingly, but there is still a lack of data, especially from the field. The authors present not only omics data, but also physiological data on an understudied region of the ocean, and nicely combine their data to reach meaningful conclusions. Thus, I generally find this study of interest to the Biogeosciences scientific community.

The article falls a bit short on some of the methods details in the meta-omics sections, which I specify further down. Figure 3 could be presented better and discussed more, as some of the results are not discussed at all.

I highly recommend this article for publication after minor revisions.

**Specific comments**

**Trichodesmium clade distribution:**

I understand that the concentrations of *Trichodesmium* cells is derived from the qPCR of *rnpB*. This should be noted in the methods part 2.2 (state that absolute numbers were derived, and briefly how it was done, rather than just referring to Rouco et al) and at the beginning of the respective results section, as well as in figure 2. In the reference given for the qPCR, Rouco et al used known cell concentrations as templates for calibration. I just want to remind the authors that *Trichodesmium* seems to be highly polyploid with hundreds of genomes per cell, and that this differed by up to factor 6 between lab cultures (which I think were used for the calibration here) and the field (Sargent et al, 2016). If the authors share my view here, I suggest discussing the resulting error briefly. In the text, I would appreciate depth-integrated cell counts for *Trichodesmium* per station, instead of averages across multiple depths and stations. I think this would be more interesting given that *Trichodesmium* vertical distribution might change within one day through vertical migration.

**Trichodesmium colony sampling**

Here, I am missing data on the speed of the net tow and the total volume filtered per tow. It is reasonable to assume that with this mesh size and a certain speed, the outer filaments of colonies are lost.

**Trichodesmium OGs from the metagenome**

The large number of OGs in the *Trichodesmium* metagenomics bin leaves me a little skeptical. While this is discussed in the paper in section 4.2, I do not think that this high number of unique OGs (9,790) "could potentially be attributed to a greater diversity of *Trichodesmium* ecotypes in the WTSP" (p17,20-21). The published *T. erythraeum* genome results in 2,982 OGs according to the authors, so this would mean a tripling of unique protein-coding gene groups within the genus *Trichodesmium* compared to *T. erythraeum*, which already is a very large genome with a wealth of unique functions, and is not known to have genomic islands that would facilitate frequent gene shuffling, like for example in *Prochlorococcus*. While this of course does not mean that this finding is necessarily incorrect, I think it might be the case that multiple OGs are indeed the same protein group, but

represent only partial proteins and are thus not grouped together in one OG when they should. To test this, I suggest mapping OG consensus sequences to the *T. erythraeum* genome with a high mismatch/gap tolerance and checking if multiple OGs often map to the same gene (especially for long genes).

**Metatranscriptomics**
Why is the expression for the microbiome in almost all KEGG categories presented in Figure 3 the lowest in sample LDB 3. This is not counterbalanced by higher transcription in the Trichodesmium bin either so it seems that reads are missing here. Could this be a normalization artefact? I understand that RSEM was used for mapping and normalization yielding TPM as normalized counts. These TPM values were additionally normalized within each OG to give all OGs equal weight. The mapping was done only to protein coding regions of the metagenomics scaffolds. Was the fraction of reads mapping to those regions similar for all samples? Maybe reads were lost to other phototrophs (not included in the metagenomics reference) or due to a very high expression of some non-coding RNAs (also not included). I would appreciate some mapping statistics to be able to evaluate this part (total and fraction of reads mapped per sample).

**KEGG submodules, Orthologous Groups (OGs), and their annotation:**
Annotating metatranscriptomes with de-novo assembled metagenome bins is always challenging. I think the idea of binning proteins into OGs, mapping the transcripts to those OG bins and then trying to classify those OGs functionally is generally a nice way to do it. I am still left a little confused with the overall process. A Diamond search against UniRef90, KEGG annotation, and clustering into OGs are mentioned as three separate annotation methods (."… annotation was also carried out using …", "Proteins … were also clustered …"). I would thus greatly appreciate a workflow diagram as a Suppl. Figure that answers the following questions: Where/How exactly did you derive the functional classification of a protein and an OG? How does this relate to the KEGG module expression profiles used in Figure 3?
The KEGG categories presented in Figure 3 are mostly subcategories of parent categories named PATHWAY MODULE, STRUCTURAL COMPLEX, FUNCTIONAL SET, and SIGNATURE MODULE, but the links to these high-level parent categories are missing, which makes some of the presented subcategories hard to interpret (the hierarchy can be found here: http://www.genome.jp/kegg-bin/get_htext#C56). Additionally, the submodule "metabolic capacity" has no real meaning unless the reader knows what metabolic capacities are lumped together within this term. Other categories also presented in Figure 3 do not appear within this hierarchy at all (e.g., sugar metabolism, phenylpropanoid and flavonoid biosynthesis). Where were they derived from? If they are children categories, why were only a few of this hierarchy presented? Please clarify and note all parent categories in figure 3. It also seems dubious that *Trichodesmium* has no proteins in the category "carbohydrate metabolism". I would double check that.
I thus suggest a revision of the mapping to KEGG categories, deleting those that are neither meaningful nor mentioned from the figure (like "drug resistance" or "metabolic capacity"), stating the parent hierarchy and a general clarification to the reader.

**Phosphonate synthesis gene cluster**
This is a nice finding. To make it bulletproof that this assembled scaffold indeed originated from a *Trichodesmium* genome, I suggest to add a suppl. figure that shows an alignment of the scaffold with all raw reads that cover the transition from the non-coding region of the *T. erythraeum* IMS101 genome to the ppm gene. Please also provide the sequence of the scaffold in a suppl. file.

**Technical corrections**

**P1, 25ff:** Split into two sentences to facilitate reading.

**P1, 29-30**: I find this sentence too generic.

**P3, 7:** add: taxonomically conserved *across ocean basins*?

**P4, 14:** "Nutrient analyses were collected" – change to *Samples for nutrient analyses were collected*

**P5, 9:** (qP**RC**) should be (qP**CR**)

**P5, 17:** "gentle vacuuming" – How gentle? Please state the mbar pressure if possible.

**P6, 1:** please clarify what this means: "after some sets of *Trichodesmium* incubations"

**P6, 13**: Please state how many colonies per sample were extracted.

**P7, 3**: What about associated phototrophs? Were those sequences ignored for all further analysis?

**P7, 17ff:** Does this mean: For the majority of all successfully annotated proteins within each orthologous group, the UniRef and KEGG annotations matched? Please re-write the annotation method to make a little clearer (see also Specific Comment section above).

**P8, 6**: Add "Maximum likelihood *phylogenetic* analysis"

**P8, 15**: I assume RNA was also extracted from isolated *Trichodesmium* colonies, but please state this here,a nd also mention how many colonies per sample were extracted.

**P8, 25**: Please state fragmentation method (ultrasound or chemical, if any), resulting fragment size, HiSeq instrument version, and specify read length.

**P9, 12**: "tags per million" should be *transcripts per million (TPM).* Please correct throughout the manuscript.

**P 10, 1-2**: I cannot find stations LDB1-3 mentioned anywhere. Figure 1 only shows one station: LDB. Maybe the authors mean *3 individual samples* from station LDB? Please see also my comment on figure 1.

**P10, 10ff**: With this sentence, the authors start using **cells/mL** for the rest of the paragraph, instead of the **cells/L** that is used before and in figure 2a. This should be corrected.
In line 11, 9,000 – 58,000 cells mL-1 are given as a concentration range for "stations west of 170°W and LDB". However, looking at figure 2a, there are clearly concentrations above 100,000 cell/L (orange; assuming units should be the same). I am not sure whether I may be reading the figure wrongly or this is due to the ODV extrapolation, or simply a mistake.
Also, to clarify east/west for the reader across the 180° longitude and not confuse it with the x-axis labels, I suggest to add in line 11: ("stations west*/left* of 170°W and LDB")

**P11:** Both sections are titled "**3.3 …**"

**P12, 8-10:** I do not see this statement reflected in figure 3.

**P18, 8**: Which one is the "broadest KEGG annotation level"? Please name it.

**P22, 15ff**: Were those phosphate reduction rates in the western North Atlantic not explainable so far? If yes, I would state this to help the reader understand this sentence better. I would also move this sentence to the end of the paragraph and re-write it. It feels a little fuzzy ("may in part underpin").

P22, 28: add "in cultured Trichodesmium isolates from Clade III, *specifically T. erythraeum,* (Dyhrman et al, 2009)"

**Figure 1:** Not clear to me which ones are the SD and which ones the LD stations. "SD" does not appear in the figure, but caption says "denoted SD". Please make that clear in the figure or text. Please increase font size of the station tags.

**Figure 2**: Also, station LDB is indicated, but has no data points. Is this correct?
It would be nicer if the panels a, b, and c all had the same x-axis, so that all the stations are indeed directly beneath each other. The difference seems to be that the x-axis in panel a is not linear.

**Figure 3**: Is the color code indicating relative expression (low to high) linear or log? I find it a bit difficult to find the functions mentioned in the Results part (p12, 5ff) in the Figure. I suggest to add a number to all categories in the figure and refer to those numbers in the text instead of just writing for example "lipid, vitamin, glycan, .. metabolism". That way the reader can easily find what the text refers to. Also, I would change the color code for the KEGG modules, and, as mentioned in the comment above, indicate the full KEGG hierarchy, and get rid of non-telling categories.

**Figure4:** The asterisks in LDB** are not mentioned in the caption. What do they mean?

**References**
Sargent, E.C., Hitchcock, A., Johansson, S.A., Langlois, R., Moore, C.M., LaRoche, J., Poulton, A.J., and Bibby, T.S. (2016). Evidence for polyploidy in the globally important diazotroph Trichodesmium. FEMS Microbiology Letters fnw244.

---

## Referee Comment (RC2) · Anonymous Referee #2 · 15 May 2018

This is an impressive dataset that has a high potential to offer tantalizing insight into the gene expression of the trichodesmium holobiont in a relatively understudied environment. In spite of my low rating, I actually think it is not far from living up to this potential. That said, I think there is some remaining work that needs to be done with respect to analysis and presentation of these data.

Minor Technical Issues: What methods were used for the biogeochemical measurements? Which of the Outpace articles are these data originally presented? Note: the link to the data from the cruise requires a login and there is no indication in this manuscript as to where these data are published (if they are). If they are not published

in the other Outpace manuscripts, the methods should be clearly presented here. If they are in other manuscripts, make it clear which manuscript contains the relevant information. What were the methods for the phosphate turnover measurements? I assume these are only presented in this manuscript. While the phosphate uptake measurements are given, the phosphate turnover details are left out.

Results Interpretation: What is different about the 3 samples from LDB? Were they different times? Different colonies? Were they supposed to be replicates? It seems there is as much if not more variability in these three samples as is seen in samples from the other stations, especially with respect to the microbiome, but also with respect to the Trichodesmium expression results. This needs to be addressed somewhere in the manuscript.

What is known about the overall expression levels of the genes that are discussed as marker genes? In other words, is the fact that these genes are among the highest expressed a sign that they are upregulated or because they are constitutively expressed at a high level? The statement on p17 that marker genes are not detectable in cultures grown under replete conditions is false. At least in the citations listed it appears the marker genes are detectable (though significantly downregulated) in replete conditions. Can you use the expression of a housekeeping gene to normalize results in some way? The mix of iron response genes listed are concerning as high expression of all of them is not actually suggestive of iron limitation. Yes, the are all linked to iron metabolism, but previous work has shown that some are upregulated and others are down regulated in situations of iron limitation, and some have shown inconsistent results. For example, Polyvou et al 2018 found that bfr, ftn, fur3, and nifH gene expression was not regulated in response to iron. While examining protein expression, not transcripts, Snow et al 2015 found that ftn was only abundant under iron replete conditions (and absent under iron limitation) and nifH was similarly higher under iron replete conditions as would be expected.

Actually, the high nifH expression and high rates of nitrogen fixation measured on the

cruise as referenced in the other articles in this special issue would suggest that maybe the Trichodesmium are doing ok. It's very hard to assess the level of iron-limitation, phosphorus-limitation, or co-limitation of the Trichodesmium based on the transcriptome data without some relative measure or other metric to assess what these expression levels mean. It seems as though the authors have potentially relevant data that they could mine to address these issues.

One of the most exciting bit about the paper is the presence of the Trichodesmium gene cassette that appears to allow the organism to perform phosphonate biosynthesis. Right now that exciting result is lost in the weeds of a convoluted discussion of micro-nutrient limitation.

Additionally, in the conclusions the authors suggest that the variable limitation could be influencing growth and nitrogen fixation. It seems that there's a lot of data from this cruise they could examine to see if this is the case. It would be nice to see them include some concrete statements referencing the other manuscripts from the cruise.

---

## Author Comment (AC1) · 29 Jun 2018

**Response to reviewers**

**\*\*\*Revised draft of manuscript and figures follows. Substantial changes are highlighted in yellow.\*\*\***

5 **Reviewer #1**

**General comments**
The article Trichodesmium physiological ecology and phosphate reduction in the western Tropical South Pacific by Frischkorn et al. is one of a series of articles on the scientific results of the OUTPACE project, a field study
10 focused on the undersampled Western Tropical South Pacific. The authors used a combination of environmental metagenomics, metatranscriptomics, nutrient analyses, and analysis of Trichodesmium holobiont phosphate physiology to shed some light on the physiological interactions between and combined metabolic potential of Trichodesmium and its associated microbiome. The study focuses on phosphorus physiology, and touches on iron limitation as well.

I find the article generally very well written and easy to follow. The results presented are relevant to the understanding of how Trichodesmium and its microbiome shape nutrient availability in the regions where they thrive. Studies on Trichodesmium as a holobiont are being published increasingly, but there is still a lack of data, especially from the field. The authors present not only omics data, but also physiological data on an understudied
20 region of the ocean, and nicely combine their data to reach meaningful conclusions. Thus, I generally find this study of interest to the Biogeosciences scientific community.

The article falls a bit short on some of the methods details in the meta-omics sections, which I specify further down. Figure 3 could be presented better and discussed more, as some of the results are not discussed at all.

I highly recommend this article for publication after minor revisions.

**Specific comments**

30 **Trichodesmium clade distribution:**
I understand that the concentrations of Trichodesmium cells is derived from the qPCR of rnpB. This should be noted in the methods part 2.2 (state that absolute numbers were derived, and briefly how it was done, rather than just referring to Rouco et al) and at the beginning of the respective results section, as well as in figure 2. In the reference given for the qPCR, Rouco et al used known cell concentrations as templates for calibration. I just want
35 to remind the authors that Trichodesmium seems to be highly polyploid with hundreds of genomes per cell, and that this differed by up to factor 6 between lab cultures (which I think were used for the calibration here) and the field (Sargent et al, 2016). If the authors share my view here, I suggest discussing the resulting error briefly.

At the reviewer's suggestion, we have updated the manuscript methods to include a description of the qPCR
40 methodology. In the discussion section we also now include a caveat pointing out the potential for polyploidy in *Trichodesmium* and how this might influence interpretation of our results. In short, the counts presented in this paper are tabulated using a standard curve generated from cell counts performed on cultures of Clade I and Clade III (Rouco et al., 2014, 2016). This analysis approach yields $C_T$ values that take into account polyploidy, unlike gene standard approaches, but the method would be prone to variation in the cell calculation if there were
45 discrepancies between ploidy levels in the field and culture for example. This is acknowledged in the revised text. To be conservative we do not draw conclusions based on the absolute cell number, only the patterns. We

note we identified trends in relative abundance that were consistent with the patterns observed with other methods employed on the cruise transect (e.g. Stenegren et al., 2017).

In the text, I would appreciate depth-integrated cell counts for Trichodesmium per station, instead of averages across multiple depths and stations. I think this would be more interesting given that Trichodesmium vertical distribution might change within one day through vertical migration.

10 The text in this section has been edited to clarify our main points about the *Trichodesmium* distribution. We agree that vertical migration of *Trichodesmium* over day-night cycles is an interesting facet of this organism's ecology, and in the revised version of the manuscript we now include a supplemental table that includes the qPCR counts from all stations and all depths, thus giving interested parties the option to view or use these results in future studies. We have edited the text in this section of the results to better reflect the depths and locations where 15 samples were obtained when specific cell concentrations are highlighted. Considering the scope of our study, the fact that *Trichodesmium* sampling occurred at roughly the same time each day, that biomass was concentrated in the upper 20 m across the entire transect, and that net tow sampling used for all other analyses captures colonies within this 20 m range, in subsequent analyses where cell count values are used, we have continued to use average concentration from the top 20 m, which includes 3 discrete depths captured by the Rosette sampling 20 device.

**Trichodesmium colony sampling**
Here, I am missing data on the speed of the net tow and the total volume filtered per tow. It is reasonable to 25 assume that with this mesh size and a certain speed, the outer filaments of colonies are lost.

We now include the time it took to perform the net tows (15 minutes total, per sample). We have also edited the text to clarify that these net tows were done by hand (rather than being towed behind the boat) and were performed gently while the ship was stopped to avoid shearing colonies. On hand net tows such as this, we do not 30 include a flow meter and we are unable to determine the exact volume filtered per tow, but we now note in the manuscript that each net tow likely filters thousands of liters of water. We perform these net tows similarly across all field studies, and so the sampling here is consistent with previous work that has consistently recovered *Trichodesmium* and its microbiome (e.g. Rouco et al., 2016; Frischkorn et al., 2017, 2018).

35 **Trichodesmium OGs from the metagenome**
The large number of OGs in the Trichodesmium metagenomics bin leaves me a little skeptical. While this is discussed in the paper in section 4.2, I do not think that this high number of unique OGs (9,790) "could potentially be attributed to a greater diversity of Trichodesmium ecotypes in the WTSP" (p17,20-21). The published T. erythraeum genome results in 2,982 OGs according to the authors, so this would mean a tripling of 40 unique protein-coding gene groups within the genus Trichodesmium compared to T. erythraeum, which already is a very large genome with a wealth of unique functions, and is not known to have genomic islands that would facilitate frequent gene shuffling, like for example in Prochlorococcus. While this of course does not mean that this finding is necessarily incorrect, I think it might be the case that multiple OGs are indeed the same protein group, but represent only partial proteins and are thus not grouped together in one OG when they should. To test 45 this, I suggest mapping OG consensus sequences to the T. erythraeum genome with a high mismatch/gap tolerance and checking if multiple OGs often map to the same gene (especially for long genes).

We agree with the reviewer that the high number of unique *Trichodesmium* OGs is surprising and given that we can only really speculate on what may be driving this we have substantially edited this section of the manuscript. The *T. erythraeum* IMS101 genome (Clade III) is not representative of field populations, which are dominated in the WTSP by Clade I, hence the necessity of using metagenome assemblies and *Trichodesmium*-identified genome bins for metatranscriptome read mapping. Clustering proteins using the MCL algorithm only considers proteins longer than 70 amino acids so as to avoid partial sequences, and uses a sequence comparison approach that groups partial proteins into the same OG if there is any sequence overlap. This approach should mitigate generating multiple OGs for the same protein. We have mapped OGs back to the *T. erythraeum* IMS101 genome as suggested. Even with a high mismatch allowance, roughly 60% of the genes from *Trichodesmium*-identified genome bins do not align to the IMS101 genome. Our group has seen similar low mapping of field *Trichodesmium* samples to the IMS101 genome (Rouco et al. 2018, *ISME J*), leading us to believe that this is not an assembly error but instead accurately reflects divergences between the dominant clades in the environment and the Clade III *T. erythraeum* represented by IMS101. Furthermore, examination of some of the key genes discussed in the paper does not show multiple OGs annotated as the same gene. This suggests that the number of OGs is not primarily driven by an artifact of the clustering. As highlighted above, the discussion of the apparent OG expansion has been heavily edited in the revised text.

**Metatranscriptomics**

Why is the expression for the microbiome in almost all KEGG categories presented in Figure 3 the lowest in sample LDB 3. This is not counterbalanced by higher transcription in the Trichodesmium bin either so it seems that reads are missing here. Could this be a normalization artefact? I understand that RSEM was used for mapping and normalization yielding TPM as normalized counts. These TPM values were additionally normalized within each OG to give all OGs equal weight. The mapping was done only to protein coding regions of the metagenomics scaffolds. Was the fraction of reads mapping to those regions similar for all samples? Maybe reads were lost to other phototrophs (not included in the metagenomics reference) or due to a very high expression of some non-coding RNAs (also not included). I would appreciate some mapping statistics to be able to evaluate this part (total and fraction of reads mapped per sample).

We thank the reviewer for bringing up this aspect of the data to our attention. In brief, we reexamined this analysis and found that overall, similar numbers of metatranscriptome reads mapped to the metagenome across all stations. To address any potential variability however, we normalized the KEGG signals within each sample individually. Even after this normalization, the distribution of reads across KEGG submodules over the samples from LDB was noticeably different than the SD stations, thus when each submodule is visualized in a heatmap as rowwide averages, these samples look different. We maintain that this normalization and display is the best way to visualize changes in *Trichodesmium* and microbiome metabolism across this WTSP transect, and have added a caveat to the revised manuscript that better explains the trends in the data. Additionally, we now note in the revised text that the broad physiological shifts visualized through KEGG module expression could be symptomatic of the crashing phytoplankton bloom that took place during the time when we sampled LDB (Stenegren et al.; de Verneil et al., Valdés et al., *this issue*). An in depth analysis of such dynamics are beyond the scope of this paper (but provide an exciting avenue for inquiry in on-going work on this dataset), however in the revised text we briefly explain this situation as further rationalization for the gene expression trends.

**KEGG submodules, Orthologous Groups (OGs), and their annotation:**

Annotating metatranscriptomes with de-novo assembled metagenome bins is always challenging. I think the idea of binning proteins into OGs, mapping the transcripts to those OG bins and then trying to classify those OGs functionally is generally a nice way to do it. I am still left a little confused with the overall process. A Diamond search against UniRef90, KEGG annotation, and clustering into OGs are mentioned as three separate annotation
5   methods (."… annotation was also carried out using …", "Proteins … were also clustered …"). I would thus greatly appreciate a workflow diagram as a Suppl. Figure that answers the following questions: Where/How exactly did you derive the functional classification of a protein and an OG? How does this relate to the KEGG module expression profiles used in Figure 3?

10   We agree with the reviewer that in our initial submission our annotation pipeline was not clear, likely because the multi-pronged approach was indeed difficult to explain with text alone. Thanks to the reviewer's helpful suggestion we now include a workflow diagram in the Supplemental Figures that outlines the pipeline we followed for assembly, annotation and analysis. We have also heavily edited the methods section detailing these protocols so that with the aid of the new diagram our approach will be more clear.

The KEGG categories presented in Figure 3 are mostly subcategories of parent categories named PATHWAY MODULE, STRUCTURAL COMPLEX, FUNCTIONAL SET, and SIGNATURE MODULE, but the links to these high-level parent categories are missing, which makes some of the presented subcategories hard to interpret
20   (the hierarchy can be found here: http://www.genome.jp/kegg-bin/get_htext#C56). Additionally, the submodule "metabolic capacity" has no real meaning unless the reader knows what metabolic capacities are lumped together within this term. Other categories also presented in Figure 3 do not appear within this hierarchy at all (e.g., sugar metabolism, phenylpropanoid and flavonoid biosynthesis). Where were they derived from? If they are children categories, why were only a few of this hierarchy presented? Please clarify and note all parent categories in figure
25   3. It also seems dubious that *Trichodesmium* has no proteins in the category "carbohydrate metabolism". I would double check that. I thus suggest a revision of the mapping to KEGG categories, deleting those that are neither meaningful nor mentioned from the figure (like "drug resistance" or "metabolic capacity"), stating the parent hierarchy and a general clarification to the reader.

30   We agree with the reviewer that the way we presented the KEGG analysis in our initial submission was not as clear as it should have been. KEGG definitions were obtained from the modules within the "Pathway module" and "Structural complex" categories and the submodules within these categories (available at https://www.genome.jp/kegg-bin/get_htext?ko00002.keg). We have also edited the figure to remove submodules that are not meaningful, at the reviewer's request. This method and organization scheme was adapted from a
35   method outlined in Alexander et al. 2015 (PNAS) and in our revised manuscript reflects this more cohesive and easy to follow organization. We note finally that it is not uncommon for certain KEGG categories to not be detected, for example the absence of the "carbohydrate metabolism" category from *Trichodesmium*. We also noted the absence of this category in our previous metagenome work (Frischkorn et al., 2017, ISME J) and attribute it to the fact that KEGG is unfortunately not optimized for non-model systems or marine microbes. We
40   highlight however the presence and expression of several other submodules attributed to *Trichodesmium* in the carbohydrate and lipid metabolism module that are depicted in revised Fig. 3.

**Phosphonate synthesis gene cluster**
45   This is a nice finding. To make it bulletproof that this assembled scaffold indeed originated from a Trichodesmium genome, I suggest to add a suppl. figure that shows an alignment of the scaffold with all raw

reads that cover the transition from the non-coding region of the T. erythraeum IMS101 genome to the ppm gene. Please also provide the sequence of the scaffold in a suppl. file.

At the reviewer's request, we now include a supplemental figure that shows reads from all three metagenome samples aligned to the phosphonate biosynthesis scaffold. This new figure shows that in all three metagenomes, reads span the non-coding region of the *T. erythraeum* IMS101 genome to the ppm gene, and mate pairs also connect the IMS101 portion to the ppm gene. We also now include an additional supplemental file that includes the full sequence of the *ppm*-containing scaffold.

Technical corrections

P1, 25ff: Split into two sentences to facilitate reading.

P1, 29-30: I find this sentence too generic.

P3, 7: add: taxonomically conserved across ocean basins?

P4, 14: "Nutrient analyses were collected" – change to Samples for nutrient analyses were collected

P5, 9: (qPRC) should be (qPCR)

P5, 17: "gentle vacuuming" – How gentle? Please state the mbar pressure if possible.

P6, 1: please clarify what this means: "after some sets of Trichodesmium incubations"

P6, 13: Please state how many colonies per sample were extracted.

P7, 3: What about associated phototrophs? Were those sequences ignored for all further analysis?

P7, 17ff: Does this mean: For the majority of all successfully annotated proteins within each orthologous group, the UniRef and KEGG annotations matched? Please re-write the annotation method to make a little clearer (see also Specific Comment section above).

P8, 6: Add "Maximum likelihood phylogenetic analysis"

P8, 15: I assume RNA was also extracted from isolated Trichodesmium colonies, but please state this here,a nd also mention how many colonies per sample were extracted.

P8, 25: Please state fragmentation method (ultrasound or chemical, if any), resulting fragment size, HiSeq instrument version, and specify read length.

P9, 12: "tags per million" should be transcripts per million (TPM). Please correct throughout the manuscript.

P 10, 1-2: I cannot find stations LDB1-3 mentioned anywhere. Figure 1 only shows one station: LDB. Maybe the authors mean 3 individual samples from station LDB? Please see also my comment on figure 1.

P10, 10ff: With this sentence, the authors start using cells/mL for the rest of the paragraph, instead of the cells/L that is used before and in figure 2a. This should be corrected.

In line 11, 9,000 – 58,000 cells mL-1 are given as a concentration range for "stations west of 170°W and LDB". However, looking at figure 2a, there are clearly concentrations above 100,000 cell/L (orange; assuming units should be the same). I am not sure whether I may be reading the figure wrongly or this is due to the ODV extrapolation, or simply a mistake.

Also, to clarify east/west for the reader across the 180° longitude and not confuse it with the x-axis labels, I suggest to add in line 11: ("stations west/left of 170°W and LDB")

P11: Both sections are titled "3.3 …"

P12, 8-10: I do not see this statement reflected in figure 3.

P18, 8: Which one is the "broadest KEGG annotation level"? Please name it.

P22, 15ff: Were those phosphate reduction rates in the western North Atlantic not explainable so far? If yes, I would state this to help the reader understand this sentence better. I would also move this sentence to the end of the paragraph and re-write it. It feels a little fuzzy ("may in part underpin").

P22, 28: add "in cultured Trichodesmium isolates from Clade III, specifically T. erythraeum, (Dyhrman et al, 2009)"

Figure 1: Not clear to me which ones are the SD and which ones the LD stations. "SD" does not appear in the figure, but caption says "denoted SD". Please make that clear in the figure or text. Please increase font size of the station tags.

Figure 2: Also, station LDB is indicated, but has no data points. Is this correct? It would be nicer if the panels a, b, and c all had the same x-axis, so that all the stations are indeed directly beneath each other. The difference seems to be that the x-axis in panel a is not linear.

Figure 3: Is the color code indicating relative expression (low to high) linear or log? I find it a bit difficult to find the functions mentioned in the Results part (p12, 5ff) in the Figure. I suggest to add a number to all categories in the figure and refer to those numbers in the text instead of just writing for example "lipid, vitamin, glycan, .. metabolism". That way the reader can easily find what the text refers to. Also, I would change the color code for the KEGG modules, and, as mentioned in the comment above, indicate the full KEGG hierarchy, and get rid of non-telling categories.

Figure4: The asterisks in LDB** are not mentioned in the caption. What do they mean?

We have revised the manuscript, making all of the above edits, correcting mistakes and omissions and adding the requested data. We thank the reviewer for editing the manuscript so carefully.

References

Sargent, E.C., Hitchcock, A., Johansson, S.A., Langlois, R., Moore, C.M., LaRoche, J., Poulton, A.J., and Bibby, T.S. (2016). Evidence for polyploidy in the globally important diazotroph Trichodesmium. FEMS Microbiology Letters fnw244.

**Reviewer #2**
This is an impressive dataset that has a high potential to offer tantalizing insight into
the gene expression of the *Trichodesmium* holobiont in a relatively understudied environment.
5   In spite of my low rating, I actually think it is not far from living up to this
potential. That said, I think there is some remaining work that needs to be done with
respect to analysis and presentation of these data.

Minor Technical Issues: What methods were used for the biogeochemical measurements?
10   Which of the Outpace articles are these data originally presented? Note:
the link to the data from the cruise requires a login and there is no indication in this
manuscript as to where these data are published (if they are). If they are not published in the other Outpace
manuscripts, the methods should be clearly presented here. If
they are in other manuscripts, make it clear which manuscript contains the relevant
15   information. What were the methods for the phosphate turnover measurements? I
assume these are only presented in this manuscript. While the phosphate uptake
measurements are given, the phosphate turnover details are left out.

We agree with the reviewer that these methods should have been presented in the first draft and regret that the
20   link we provided was not available to the public. In the edited manuscript we have revised our geochemical
analyses using new data from a broader set of iron measurements (Guieu et al., 2018 *Scientific Reports,* in press)
as well as using new nM phosphate measurements detected with a low level phosphate assay provided by our co-
authors Guieu and Louis. In the revised version of our manuscript we now delineate these methods as requested,
and incorporate these data into our analyses. These values are also reflected in the revised supplemental table.

Results Interpretation: What is different about the 3 samples from LDB? Were they
different times? Different colonies? Were they supposed to be replicates? It seems
there is as much if not more variability in these three samples as is seen in samples
from the other stations, especially with respect to the microbiome, but also with respect
30   to the Trichodesmium expression results. This needs to be addressed somewhere in
the manuscript.

We appreciate the reviewer pointing out that we omitted discussing this trend in our initial submission. The
samples from LDB were collected on three different days in the same water mass, during the 6 days that the
35   expedition spent at this "long duration" station. All samples were collected at the same time each morning. We
have edited the text surrounding this LDB sampling in the revised manuscript to make this more clear.
Furthermore, in the revised text we also discuss how LDB samples were taken during a declining phytoplankton
bloom (de Verneil et al. *Biogeosci,* this issue), which might explain the high degree of variability. We discuss the
implications of this in the revised version of the paper.

What is known about the overall expression levels of the genes that are discussed as
marker genes? In other words, is the fact that these genes are among the highest expressed
a sign that they are upregulated or because they are constitutively expressed
at a high level? The statement on p17 that marker genes are not detectable in cultures
45   grown under replete conditions is false. At least in the citations listed it appears
the marker genes are detectable (though significantly downregulated) in replete conditions.

Can you use the expression of a housekeeping gene to normalize results in
some way? The mix of iron response genes listed are concerning as high expression
of all of them is not actually suggestive of iron limitation. Yes, the are all linked to
iron metabolism, but previous work has shown that some are upregulated and others
5   are down regulated in situations of iron limitation, and some have shown inconsistent
results. For example, Polyvou et al 2018 found that bfr, ftn, fur3, and nifH gene expression
was not regulated in response to iron. While examining protein expression,
not transcripts, Snow et al 2015 found that ftn was only abundant under iron replete
conditions (and absent under iron limitation) and nifH was similarly higher under iron
10  replete conditions as would be expected. Actually, the high nifH expression and high rates of nitrogen fixation
measured on the cruise as referenced in the other articles in this special issue would suggest that maybe the
Trichodesmium are doing ok. It's very hard to assess the level of iron-limitation,
phosphorus-limitation, or co-limitation of the Trichodesmium based on the transcriptome
data without some relative measure or other metric to assess what these expression
15  levels mean. It seems as though the authors have potentially relevant data that
they could mine to address these issues.

We regret that our discussion of the *Trichodesmium* expression patterns was not clear. In the revised version of
20  the paper we have added some additional analysis (Figure 4), as well as clarified, and streamlined the discussion
of this section as requested. With the feedback from the reviewer, we have removed what was initially Figure 4a
from the analysis.  Although the high expression of many of our gene targets was tantalizing, the reviewer is
correct that we can't confirm that this is driven by limitation scenarios without a comparison to a more replete
condition.  Rather, we focus our discussion on relative patterns of expression across the SD stations and the LD
25  time-series. The focus on relative OG expression patterns across the sample set did not necessitate normalization
to a reference gene, and our method for this analysis approach (Frischkorn et al., 2018, *ISME J*) is clarified in the
revised text.

In the revised manuscript, we have limited our nutrient responsive gene analysis to only those that have been
30  experimentally shown to respond with significantly increased transcript/protein expression in response to
limitation conditions resulting in the low P responsive gene set (*sphX, ptxD, phoX*) (Dhyrman et al., 2006;
Orchard et al., 2009; Polyviou et al., 2016), the low Fe responsive set (*fld1/2, isiA, fbaA, idiA*) (Chappell and
Webb 2010; Chappel et al., 2012; Snow et al. 2015), and the co-limitation responsive set (*aroB, metE, FMO,
OXSM1*) (Walworth et al., 2017). In our revised Figure 4, we show how that the expression of the majority of the
35  P and Fe responsive OGs track together, showing similar expression dynamics across the transect. Notable
exceptions are the expression of *phoA,* which clustered with the Fe responsive set and away from its counterpart
*phoX* (potentially because P responsive *phoX* has an Fe-Ca cofactor that has repressed expression relative to *phoA*
in low Fe environments (Rouco et al., 2018)), and *phnJ*. In the revised analysis we include only Fe and P OGs
that track together across stations in the K-S testing to evaluate significant enrichment patterns (Figure 4b).  The
40  results of this new clustering show an enrichment in Fe responsive gene expression at SD2, an enrichment in P
responsive gene expression at the first day from LDB, and mirroring in overall patterns of enrichment for low P
responsive and co-limitations responsive gene sets.   Because expression of these genes/proteins are
experimentally verified to increase during limitation, and because of the similar expression patterns previously
discussed, we feel that the patterns and dynamics of these sets—rather than the overall high expression, which we
45  no longer discuss—are potent markers of the physiological state of *Trichodesmium* in a typically under-sampled
region in the oligotrophic ocean, thus adding valuable insight to this organism's ecology and physiology.

We have also edited the text surrounding the discussion of *nifH*—in the revised discussion and analysis we now show that the expression of this iron requiring enzyme subunit clusters with the Fe-marker genes, a potential indicator of the tight modulation of cellular Fe demand in *Trichodesmium*. We are unable to directly compare specific *Trichodesmium nifH* signals with measurements of bulk water column $N_2$ fixation, however both the data presented in the revised text, and studies from the special issue, consistently suggest that Fe and P are important drivers of phytoplankton physiology in this region of the WTSP. The manuscript now better integrates OUTPACE papers and their findings into the discussion.

One of the most exciting bit about the paper is the presence of the Trichodesmium
gene cassette that appears to allow the organism to perform phosphonate biosynthesis.
Right now that exciting result is lost in the weeds of a convoluted discussion of
micro-nutrient limitation.

At the reviewer's request we have streamlined the results and discussion of earlier sections of the paper (see comments above).

Additionally, in the conclusions the authors suggest that the variable limitation could
be influencing growth and nitrogen fixation. It seems that there's a lot of data from this
cruise they could examine to see if this is the case. It would be nice to see them include
some concrete statements referencing the other manuscripts from the cruise.

In the revised text we now include a more integrated discussion referencing other studies to put our observations from the WTSP into greater context. We note however that the other samples discussed in this special issue were taken from bulk water rather than the method we employed of isolating low density *Trichodesmium* colonies. Therefore, measurements of $N_2$ fixation and other parameters reflect free living planktonic microbes and are not directly comparable to genomic, gene expression, or activity data that we present in this manuscript.

[revised manuscript text omitted]

**Supplemental /tables**

**Supplemental Table 1.** Summary of water column (10 m) geochemical data collected across the OUTPACE cruise transect. *Measurement obtained at 30 m.

| Station | Lat | Lon | Water colum dissolved inorganic phosphorus (nmol/L) at 10 m | | Water column phosphate turnover (1/hr) | | Phosphate turnover time (hrs) | Water column phosphate uptake rate (nmol/L/hr) | Water column iron (nmol/L) at 10 m | Tricho. colony: % of PO4 uptake as MPn | | Tricho. colony: % PO4 uptake as PO3 | | Sum MPn PO3 (%) |
|---|---|---|---|---|---|---|---|---|---|---|---|---|---|---|
| | | | avg. | stdev | avg | stdev | | | | avg | stdev | avg | stdev | |
| SD1 | -18.0 | 159.9 | 11.2 | 0.5376 | | | | | 0.67 | | | | | |
| SD2 | -18.6 | 162.1 | 16.4 | 0.7872 | | | | | 0.36 | 0.83 | 0.12 | 1.05 | 0.33 | 1.87 |
| SD3 | -19.5 | 165.0 | 8.3 | 0.3984 | 0.0251 | 0.0013 | 39.7986 | 0.209 | 0.38 | | | | | |
| LDA | -19.2 | 164.6 | 13.8* | 0.6624 | 0.0494 | 0.0031 | 20.2525 | 0.681 | 0.85 | 0.55 | 0.12 | 0.55 | 0.00 | 1.10 |
| SD4 | -20.0 | 168.0 | 4.8 | 0.2304 | | | | | 0.69 | | | | | |
| SD5 | -22.0 | 170.0 | 2.3 | 0.1104 | 0.0026 | 0.0004 | 383.4898 | 0.006 | 0.44 | | | | | |
| SD6 | -21.4 | 172.1 | 13.4 | 0.6432 | | | | | | | | | | |
| SD7 | -20.8 | 174.3 | 10.21 | 0.49008 | 0.0029 | 0.0000 | 340.9558 | 0.030 | 0.21 | | | | | |
| SD8 | -20.7 | 176.4 | 8.2 | 0.3936 | 0.0012 | 0.0009 | 827.7212 | 0.010 | 0.38 | | | | | |
| SD9 | -21.0 | 178.6 | 6.1 | 0.2928 | 0.0389 | 0.0004 | 25.7207 | 0.237 | 0.22 | 0.58 | 0.14 | 0.77 | 0.005 | 1.35 |
| SD10 | -20.5 | -178.5 | 6.6 | 0.3168 | | | | | 0.97 | | | | | |
| SD11 | -20.0 | 175.7 | 6.6 | 0.3168 | 0.0366 | 0.0002 | 27.3563 | 0.241 | 1.16 | 0.34 | 0.00 | 0.55 | 0.01 | 0.88 |
| SD12 | -19.5 | 172.8 | 18.7 | 0.8976 | 0.0019 | 0.0008 | 519.0585 | 0.036 | 0.94* | | | | | |
| LDB | -18.2 | 170.7 | 7 | 0.336 | 0.0694 | 0.0044 | 14.4041 | 0.486 | 0.65 | 1.20 | 0.28 | 0.80 | 0.38 | 2.00 |
| SD13 | -18.2 | 169.1 | | | | | | | | | | | | |
| LDC | -18.4 | -165.9 | | | 0.5041 | 0.2785 | 1.9839 | | 0.37 | | | | | |
| SD14 | -18.4 | -163.0 | 214 | 10.272 | | | | | 0.46 | | | | | |
| SD15 | -18.3 | -160.0 | 230 | 11.04 | | | | | 0.31 | | | | | |

**Supplemental Table 2.** qPCR counts of *Trichodesmium* Clade I and Clade III obtained across the
transect.

| Station # | Clade I (VI-I type) | | | Clade III (IMS-101 type) | | |
|---|---|---|---|---|---|---|
| | Depth (m) | Cells/Liter | SD | Depth (m) | Cells/Liter | SD |
| 2.1 | 105.33 | 49.22 | 3.54 | 105.33 | 0.00 | 0.00 |
| 2.2 | 70.32 | 256.52 | 46.19 | 70.32 | 0.00 | 0.00 |
| 2.3 | 35.88 | 16935.72 | 3178.65 | 35.88 | 4241.04 | 187.77 |
| 2.4 | 15.45 | 78277.16 | 11013.84 | 15.45 | 34467.80 | 2706.92 |
| 2.5 | 8.53 | 40360.93 | 6311.34 | 8.53 | 17495.42 | 1212.01 |
| 2.6 | 5.45 | 78681.67 | 1277.21 | 5.45 | 21452.43 | 2785.86 |
| 4.1 | 106.42 | 63.98 | 8.21 | 106.42 | 0.00 | 0.00 |
| 4.2 | 69.98 | 1222.95 | 285.07 | 69.98 | 0.00 | 0.00 |
| 4.3 | 35.46 | 12498.82 | 1218.82 | 35.46 | 1457.31 | 183.75 |
| 4.4 | 16.73 | 11120.73 | 1051.08 | 16.73 | 5596.86 | 161.16 |
| 4.5 | 9.64 | 44795.69 | 8852.79 | 9.64 | 28478.09 | 4533.35 |
| 4.6 | 4.30 | 24450.06 | 4167.23 | 4.30 | 22152.05 | 1674.11 |
| 6.1 | 119.68 | 581.66 | 154.77 | 119.68 | 0.00 | 0.00 |
| 6.2 | 79.68 | 1456.09 | 99.28 | 79.68 | 391.01 | 49.60 |
| 6.3 | 40.70 | 6462.53 | 376.48 | 40.70 | 4515.22 | 99.35 |
| 6.4 | 18.17 | 40658.71 | 8811.30 | 18.17 | 13284.30 | 1015.35 |
| 6.5 | 11.23 | 149154.78 | 10230.38 | 11.23 | 22935.19 | 3787.89 |
| 6.6 | 5.38 | 82856.35 | 13163.61 | 5.38 | 23110.81 | 6456.73 |
| 9.1 | 134.66 | 108.49 | 15.83 | 134.66 | 0.00 | 0.00 |
| 9.2 | 90.08 | 44.12 | 10.07 | 90.08 | 0.00 | 0.00 |
| 9.3 | 45.18 | 5761.77 | 1220.23 | 45.18 | 2483.23 | 194.07 |
| 9.4 | 21.53 | 4628.82 | 1255.19 | 21.53 | 3010.94 | 171.89 |
| 9.5 | 12.52 | 6162.76 | 345.63 | 12.52 | 3067.23 | 136.50 |
| 9.6 | 4.331 | 20587.22 | 717.25 | 4.331 | 9164.47 | 2635.38 |
| 11.1 | 135.83 | 14.17 | 4.03 | 135.83 | 0.00 | 0.00 |
| 11.2 | 89.59 | 72.95 | 4.20 | 89.59 | 0.00 | 0.00 |
| 11.3 | 46.07 | 2152.23 | 383.28 | 46.07 | 1153.82 | 55.61 |
| 11.4 | 21.39 | 20241.08 | 1360.48 | 21.39 | 18049.39 | 1381.87 |
| 11.5 | 12.35 | 12213.64 | 770.56 | 12.35 | 8402.08 | 1165.03 |
| 11.6 | 5.84 | 36343.33 | 1861.80 | 5.84 | 28713.60 | 5077.47 |
| 12.1 | 127.48 | 13.34 | 4.36 | 127.48 | 0.00 | 0.00 |

| | | | | | | |
|------|--------|----------|---------|--------|---------|---------|
| 12.2 | 84.81 | 114.12 | 7.08 | 84.81 | 0.00 | 0.00 |
| 12.3 | 43.39 | 24045.90 | 1552.06 | 43.39 | 4784.42 | 327.22 |
| 12.4 | 19.16 | 15179.65 | 2300.00 | 19.16 | 5259.23 | 47.69 |
| 12.5 | 10.49 | 10852.23 | 2110.81 | 10.49 | 6513.01 | 1419.06 |
| 12.6 | 5.75 | 2990.12 | 366.52 | 5.75 | 2011.18 | 386.35 |
| 14.1 | 165.17 | 56.86 | 0.00 | 165.17 | 0.00 | 0.00 |
| 14.2 | 110.71 | 107.76 | 23.75 | 110.71 | 0.00 | 0.00 |
| 14.3 | 56.22 | 89.37 | 11.96 | 56.22 | 0.00 | 0.00 |
| 14.4 | 26.26 | 70.62 | 8.46 | 26.26 | 0.00 | 0.00 |
| 14.5 | 15.30 | 86.11 | 33.82 | 15.30 | 0.00 | 0.00 |
| 14.6 | 6.01 | 72.03 | 12.37 | 6.01 | 0.00 | 0.00 |
| 15.1 | 174.16 | 14.62 | 9.37 | 174.16 | 0.00 | 0.00 |
| 15.2 | 115.72 | 15.80 | 3.85 | 115.72 | 0.00 | 0.00 |
| 15.3 | 58.03 | 13.93 | 0.11 | 58.03 | 0.00 | 0.00 |
| 15.4 | 26.58 | 65.20 | 11.48 | 26.58 | 0.00 | 0.00 |
| 15.5 | 16.81 | 73.24 | 9.42 | 16.81 | 0.00 | 0.00 |
| 15.6 | 5.44 | 67.07 | 1.25 | 5.44 | 0.00 | 0.00 |

**Supplemental Table 3.** Table of gene expression counts (tags per million, TPM) summed across orthologous groups (OGs) and partitioned between the *Trichodesmium* and microbiome fractions.

**Supplemental Table 4.** The sequence of the assembled scaffold containing the *ppm* cassette from the *Trichodesmium* metagenome assembly. Note: full sequence will be included at re-submission.

```
>scaffold_3391
ATTTTGTAACCCTTTTTTCTGCTATCCCTTTGCTCCATCTCCTGACTCGTGACTCCTGTACCTCGCCCTA
CTGACGGCTCCCCTACCCGATCAAAAAACTTAATAGTGGCTATTGATACCAAATGGGTTCTATAGAGTTG
GAAGTTGTTGAATGAATTTTTGGTAGTCCATTTTGGTCTGTTGGTTCTTTGGGGTTATTGTAGAGGTTGG
TTGAGTGGGGGGCGATCGTTTTTGTGGGGAAATGGGGAATAATTAGAGGTGTTAAAGTTTTCTACTCCCT
TCCCGGAGTAATACATTCAGTATAATTATAATTATGTCCGTTAACAAGATAAGTTTATGCTCAGAAAAAC
AACCCAATTGAAGAACTTACTGCAGTCAGAAAAATTGGAATTTTTGATGGAAGCTCATAACGGCTTGAGT
GCAAAAATTGTAGAAGAAGCAGGCTTTAAGGGCATTTGGGGGAGCGGTCTTTCAATTTCGGCAGCAATGG
GTGTCAGGGATAATAATGAAGCTAGCTGGACTCAGATCTTAGAAATTCTGGAGTTTATGTCTGATGCTAC
CTCAATTCCTATTTTACTAGATGGTGACACGGGTTATGGTAACTTTAATAACTTACGACGTTTGGTAAAA
AAACTAGAGCAGCGTGGTGTGGCAGGAGTTTGTATTGAAGATAAACAGTTTCCGAAAAAAAAATAGTTTTA
TTGACGGTCATACTCAACCACTAGCTGATATAGAAGAGTTTTGTGGCAAAATCAAAGCAGCAAAAGATGC
TCAAAAGGATAATGATTTTGTAGTAATTTCTCGTGTAGAAGCTTTTATTGCCGGTTGGGGTTTGTCAGAA
GCACTAAAACGGGCAGAAGCATATTATCATTCCGGAGCTGACGGAATTTTGATTCACAGTTCTCTAAGAG
TTCCAAATGAAATTTTGGCATTTAAACAAGATTGGGGCGATCGTTGTCCTGTAATTATTGTGCCGACTAA
GTATTATACTACTCCCACTCAAGTATTTAGAGATTACAAGTTCTCAATTGCTATTTGGGCCAATCAAACT
CTAAGGGCGGCGATCACAGAAATGCAGAAAATAGTTAATAAATTATCGCGACAAGAAAATTTACTGGATG
CAGAAGAGTCAATTATTCCTGTTTCTGAGGTTTTTAGACTTCAAGGTTGCCTTGAGTTACAAGAAGCAGA
AAAACTTTATTTACCCCAAAATAACAAAAAAATAGCAAGTTTATTGTTAGCAGCATCTAGAGGTATAGAA
TTAGGAACTTTTACTAAAGAAAAACCTAAATGTATGTTGACTTTAAAAAGAAAGCCGATTTTAGGACAAA
TAATAGCTATTCATCATGAAATAGGAATAAAAAAATATTAGTGTTGTTCGCGGTTATAAAAAAGAAGCTAT
TAATTTAGCCAACATTAAATATGTAGATAACGATGAATATGAGTCAACGGGGGAAATATTTTCCCTCTAT
TTAGGCTTGAAAAAAAATCATTCAAGATAATGAAAATCAAGACCTAGTAATTGGCTATGGTGATGTGTTAT
TGAAAAAGTATATACTGCAACTCCTCTTAGAAAATAATTATGACTTGGCAGTAATAGTTGATAGCAACTG
GCATAAATGCTCGAATCAAGTTAGGGCAGATTATACTAATTGCTCTGTACCTAATTCTAAACGAGCATTT
TATCAAGATGTTGGCTTACACCAAATAGACGCAAATATGCCAGAAGAAAAAATTTGTGGAATGTGGACTG
GTTTACTGAAAGTTTCAAGTAACGTGCAAAGACCATTATTAGATGCTCTTGAAAAGTTACTTCTACAGCT
GGAAATCAAAAGTAGCGGTAGAATGCCAAACTTGATAAATGAGTTAATTGCTTTAGGATATCCAGTTAAT
GTGGTTTACATTACAGGAGATTGGTTAGATGTAGATGAAGTAGAAGATATGATCAAGGCGGGAACTTTTT
AAATGATTAAAGCAGAAAACTTTTTTGAAGTTGCAGGCAAACTAGGATTTGGCTTATATACAGGGGTTCC
CTGTTCTTACCTGAAGTCTTTTATAAACTATGTAATTGATTCTCCTGACTTGCGGTATGTAGGTGCTACA
AATGAGGGAAATGCGGTAGCTATCGCCTCTGGTGCAGAATTAGCCGGAGTTAGAAGTGTGGTTATGTTGC
AGAATTCTGGTTTGGGTAATGGGGTTAACCCGCTAACATCTCTGAATCAAACCTTTAAAATACCAATTTT
```

**Supplemental Figures**

**Supplemental Figure 1.** Diagram detailing approach used to assemble and partition the metagenome between *Trichodesmium* and the microbiome, annotate and cluster protein coding sequences into orthologous groups (OGs), and obtain gene expression values from metatranscriptomes. Eukaryote identified sequences as well as phototrophs other than *Trichodesmium* were excluded from downstream analysis after the genome binning and analysis step. TPM, transcripts per million.

[Figure]

**Supplemental Figure 2.** Distribution and annotations of KEGG functional annotations of OGs found uniquely in the *Trichodesmium* (T) or microbiome (M) or those composed of both *Trichodesmium* and microbiome proteins (B). (a) Total number of OGs in each category. (b) A functional breakdown of these annotations at the KEGG module level and detailed annotations from within two KEGG module categories.

[Figure]

**Supplemental Figure 3.** Metagenome reads aligned to the scaffold containing the *ppm* cassette. Thick black rectangles depict ~100 bp reads, connected by thin black lines to their mate paired read.

[Figure]

**Supplemental Figure 4.** Amino acid alignment of the Ppm protein recovered from a *Trichodesmium* metagenome bin against experimentally verified Ppm sequences in other organisms. Genes highlighted in green denote regions with 100% amino acid identity across all sequences. Shades of yellow denote conservation across the majority of amino acids in the column. Un-highlighted amino acids indicate divergent residues or regions with little conservation. Purple and grey highlighted columns denote residues that were previously determined to be important to the structure or activity of this enzyme (Chen et al., 2006).

[Figure]

**Supplemental Figure 5.** Phylogenetic tree showing the placement of the *Trichodesmium* Ppm protein (red), a microbiome proteins from this study's metagenome assembly that is similar to Ppm but lacking conservation at key residues (green), as well as homologous proteins from a previously assembled North Atlantic *Trichodesmium* metagenome assembly (blue) (Frischkorn et al., 2017), along with homologous sequences obtained from the NCBI nr database (black). The tree was generated with FastTree using the default settings (Price et al., 2010). Numbers at the branch labels indicate FastTree support percentages for the sequences in that branch.

[Figure]